# Horizon-Independent Minimax Linear Regression

**Alan Malek**
Laboratory for Information and Decision Systems
Massachusetts Institute of Technology
77 Massachusetts Avenue
Cambridge, MA 02139-4307, USA `amalek@mit.edu`

**Peter L. Bartlett**
Department of EECS and Statistics
University of California
Berkeley, CA 94720-1776, USA
`bartlett@cs.berkeley.edu`

## Abstract

We consider online linear regression: at each round, an adversary reveals a covariate vector, the learner predicts a real value, the adversary reveals a label, and the learner suffers the squared prediction error. The aim is to minimize the difference between the cumulative loss and that of the linear predictor that is best in hindsight. Previous work demonstrated that the minimax optimal strategy is easy to compute recursively from the end of the game; this requires the entire sequence of covariate vectors in advance. We show that, once provided with a measure of the scale of the problem, we can invert the recursion and play the minimax strategy without knowing the future covariates. Further, we show that this forward recursion remains optimal even against adaptively chosen labels and covariates, provided that the adversary adheres to a set of constraints that prevent misrepresentation of the scale of the problem. This strategy is horizon-independent in that the regret and minimax strategies depend on the size of the constraint set and not on the time-horizon, and hence it incurs no more regret than the optimal strategy that knows in advance the number of rounds of the game. We also provide an interpretation of the minimax algorithm as a follow-the-regularized-leader strategy with a data-dependent regularizer and obtain an explicit expression for the minimax regret.

## 1 Introduction

Linear regression is a fundamental prediction problem in machine learning and statistics. In this paper, we study a sequential version: on round $t$, the adversary chooses and reveals a covariate vector $\boldsymbol{x}_t \in \mathbb{R}^d$, the learner makes a real-valued prediction $\hat{y}_t$, the adversary chooses and reveals the true outcome $y_t \in \mathbb{R}$, and finally the learner is penalized by the square loss, $(\hat{y}_t - y_t)^2$.

Since it is hopeless to guarantee a small loss (the adversary can always cause constant loss per round), we instead aim to guarantee that we are able to predict almost as well as the best fixed linear predictor in hindsight. Letting $\boldsymbol{x}_s^t$ and $y_s^t$ denote $\boldsymbol{x}_s, \ldots, \boldsymbol{x}_t$ and $y_s, \ldots, y_t$, respectively, define the regret of a strategy that predicts $\hat{y}_1^T$ as

$$\mathcal{R}_T\left(\hat{y}_1^T, \boldsymbol{x}_1^T, y_1^T\right) := \sum_{t=1}^{T}(\hat{y}_t - y_t)^2 - \min_{\theta \in \mathbb{R}^d}\sum_{t=1}^{T}(\theta^\top \boldsymbol{x}_t - y_t)^2.$$

A strategy $s : \bigcup_{t \geq 1} (\mathbb{R}^d \times \mathbb{R})^{t-1} \times \mathbb{R}^d \to \mathbb{R}$, is a map from observations to predictions, and we define $\mathcal{R}_T \left( s, \boldsymbol{x}_1^T, y_1^T \right) := \mathcal{R}_T \left( \hat{y}_1^T, \boldsymbol{x}_1^T, y_1^T \right)$ where $\hat{y}_t = s(\boldsymbol{x}_1, y_1, \ldots, \boldsymbol{x}_{t-1}, y_{t-1}, \boldsymbol{x}_t)$. Our goal is to find a strategy that guarantees low regret for all data sequences. In particular, this paper is concerned with the minimax strategy $s^*$, which is the strategy that minimizes the worst case regret over all possible covariate and outcome sequences in some constraint set, i.e. $s^*$ satisfies

$$\max_{\boldsymbol{x}_1^T, y_1^T} \mathcal{R}_T \left( s^*, \boldsymbol{x}_1^T, y_1^T \right) = \min_s \max_{\boldsymbol{x}_1^T, y_1^T} \mathcal{R}_T \left( s, \boldsymbol{x}_1^T, y_1^T \right).$$

In general, computing minimax strategies is computationally intractable because the optimal prediction $\hat{y}_t$ depends on the complete history $(\boldsymbol{x}_1, y_1, \ldots, \boldsymbol{x}_{t-1}, y_{t-1}, \boldsymbol{x}_t)$, and the dependence might be a rather arbitrary function of this enormous space of histories. So it is surprising that, in the case of fixed-design linear regression (where the strategy knows the covariate sequence in advance), the minimax strategy can be efficiently computed [Bartlett et al., 2015].

This paper builds on results from Bartlett et al. [2015], which studied fixed-design online linear regression, where the game length $T$ and covariates $\boldsymbol{x}_1^T := \boldsymbol{x}_1, \ldots, \boldsymbol{x}_T$ are known to the learner a priori. Under constraints on the adversarial labels $y_1^T$, the value function and minimax strategy were calculable in closed form using backwards induction. The resulting minimax strategy

$$\hat{y}_{t+1} = \boldsymbol{x}_{t+1}^\top \boldsymbol{P}_{t+1} \sum_{s=1}^t y_s \boldsymbol{x}_s, \qquad \text{(MMS)}$$

is a simple, linear predictor with coefficient matrices defined by

$$\boldsymbol{P}_T = \left( \sum_{t=1}^T \boldsymbol{x}_t \boldsymbol{x}_t^\top \right)^\dagger \text{ and recursion } \boldsymbol{P}_t = \boldsymbol{P}_{t+1} + \boldsymbol{P}_{t+1} \boldsymbol{x}_{t+1} \boldsymbol{x}_{t+1}^\top \boldsymbol{P}_{t+1}. \qquad (1)$$

The $\hat{y}_t$ is a function of the whole sequence $\boldsymbol{x}_1^T$, and thus an extension to online-design seems difficult.

**Our contributions** This paper extends the fixed design setting to adversarial design where neither the covariates nor the length of the game are fixed a priori. We use $\{\boldsymbol{x}_t\}$ and $\{y_t\}$ to denote arbitrary length sequences of covariates and labels, respectively. We allow the adversary to play any covariate sequence in some constraint set $\mathcal{X}$ and labels in some set $\mathcal{Y}(\{\boldsymbol{x}_t\})$ (which may depend on the covariates).

In particular, we identify a family $\mathcal{X}, \mathcal{Y}$ parameterized by a positive-definite matrix $\boldsymbol{\Sigma}$, representing the size of future covariates, and a scalar $\gamma_0$, representing the size of the future labels, and present a strategy that is minimax optimal against all adversarial sequences in this family. The algorithm needs only know $\boldsymbol{\Sigma}$, and the guarantee is horizon-independent in the sense that the family does not constrain the length of the covariate sequence and includes covariate sequences of arbitrary length for any $\boldsymbol{\Sigma}, \gamma_0$ pair.

Given: covariate constraints $\mathcal{X}$ and label constraints $\mathcal{Y}(\{\boldsymbol{x}_t\})$
For $t = 1, 2, \ldots,$

- Adversary chooses $\boldsymbol{x}_t$ s.t. $\boldsymbol{x}_1^t \in \mathcal{X}$
- Learner predicts $\hat{y}_t$
- Adversary may end the game
- Adversary reveals $y_t$ s.t. $y_1^t \in \mathcal{Y}(\boldsymbol{x}_1^T)$
- Learner incurs loss $(\hat{y}_t - y_t)^2$
- The game ends if no $\boldsymbol{x}_{t+1}$ exists such that $\boldsymbol{x}_1^{t+1} \in \mathcal{X}$

Figure 1: Adversarial Covariates Protocol

The protocol of the general, horizon-independent setting is outlined in Figure 1. We derive the minimax strategy and show that it is optimal in the following way.

**Definition 1.** *A strategy $s^*$ is horizon-independent minimax optimal for some class $\mathcal{X}$ of covariate sequences and some class $\mathcal{Y}(\{\boldsymbol{x}_t\})$ of label sequences, possibly depending on $\{\boldsymbol{x}_t\} \in \mathcal{X}$, if*

$$\sup_T \left( \sup_{\boldsymbol{x}_1^T \in \mathcal{X}, \, y_1^T \in \mathcal{Y}(\boldsymbol{x}_1^T)} \mathcal{R}_T \left( s^*, \boldsymbol{x}_1^T, y_1^T \right) - \min_s \sup_{\boldsymbol{x}_1^T \in \mathcal{X}, \, y_1^T \in \mathcal{Y}(\boldsymbol{x}_1^T)} \mathcal{R}_T \left( s, \boldsymbol{x}_1^T, y_1^T \right) \right) = 0.$$

We require $s^*$ to have regret no larger than *even a strategy that knows $T$*.

In other words, we establish a more natural measure of game length than the number of rounds. The covariate constraints on $\{\boldsymbol{x}_t\}$ ensure that the adversary respects the scale constraint $\boldsymbol{\Sigma}$ so that the

learner is not led to under-regularize or over-regularize. The minimax strategy is efficient and is simultaneously minimax optimal against all covariate sequences corresponding to $\boldsymbol{\Sigma}$.

We motivate our constraint set by showing that every condition is necessary, and we also cast the minimax strategy as follow the regularized leader strategy with a data-dependent regularizer. Finally, we provide a general regret upper bound.

**Outline** We begin with a review of how backwards induction is used to derive the fixed-design minimax algorithm (MMS) in Section 2. By inverting the recursion, we show in Section 3 how to calculate (MMS) given only $\boldsymbol{P}_0$, and thus we have the minimax strategy for any covariate sequence that perfectly agrees with the given $\boldsymbol{P}_0$.

Section 4 greatly expands the scope of our algorithm by deriving weaker conditions on the adversary and proves that, under these conditions, the same minimax strategy is horizon-independent minimax optimal. We argue that these conditions are necessary. We then interpret the minimax strategy as a follow the regularized leader with a specific, data-dependent regularizer in Section 5.

**Related Work** While linear regression has a long history in statistics and optimization, its online sibling is much more recent, starting with the work of Foster [1991], which considered binary labels and $\ell_1$-constrained parameters $\theta$. He proved an $O(d \log(dT))$ regret bound for an $\ell_2$-regularized follow-the-leader strategy. Cesa-Bianchi et al. [1996] considered $\ell_2$-constrained parameters and gave $O(\sqrt{T})$ regret bounds for a gradient descent algorithm with $\ell_2$ regularization. Kivinen and Warmuth [1997] showed that an Exponentiated Gradient algorithm with relative entropy gives the same regret without the need for a constraint on the parameters. Vovk [1998] applied the Aggregating Algorithm [Vovk, 1990] to continuously many experts and arrived at a scale free algorithm by using the inverse second moment matrix of past and current covariates. Forster [1999] and Azoury and Warmuth [2001] showed that this algorithm is last step minimax and achieves an $O(\log T)$ scale-dependent regret bound. (See also the work of Moroshko and Crammer [2014] on last-step minimax.)

Takimoto and Warmuth [2000] obtained the minimax strategy for prediction in Euclidean space with squared loss. This was extended to more general losses in [Koolen et al., 2014] and to tracking problems in [Koolen et al., 2015]. Finally, Bartlett et al. [2015] obtained the minimax strategy for fixed-design linear regression. We present this strategy in the next section, because we build on these results. In these papers, the minimax analysis provides a natural, data-dependent regularization, in contrast to the follow-the-leader methods described above. We make this comparison explicit in Section 5, by calculating the implied regularization.

## 2 Fixed Design Linear Regression

We begin by summarizing the main results of Bartlett et al. [2015]. Recall that in the fixed design setting, the game length $T$ and covariates $\boldsymbol{x}_1^T$ are fixed and known to both players. Define the summary statistics $\boldsymbol{s}_t := \sum_{s=1}^t y_s \boldsymbol{x}_s$, $\sigma_t^2 = \sum_{s=1}^t y_t^2$, and $\Pi_t = \sum_{s=1}^t \boldsymbol{x}_s \boldsymbol{x}_s^\top$. The minimax strategy can be computed by solving the offline problem $\min_\theta \sum_{t=1}^T (\boldsymbol{x}_t^\top \theta - y_t)^2 = \sum_{t=1}^T y_t^2 - \boldsymbol{s}_T^\top \Pi_T^\dagger \boldsymbol{s}_T$, where $\boldsymbol{M}^\dagger$ is the pseudo-inverse of matrix $\boldsymbol{M}$. The optimal actions $\hat{y}_t$ and $y_t$ are computed as a function of the state $\boldsymbol{s}_{t-1}$ and covariates $\boldsymbol{x}_1^T$ by solving the backward induction

$$V\left(\boldsymbol{s}_t, \sigma_t^2, t, \boldsymbol{x}_1^T\right) := \min_{\hat{y}_{t+1}} \max_{y_{t+1}} \left( (\hat{y}_{t+1} - y_{t+1})^2 + V\left(\boldsymbol{s}_t + y_{t+1}\boldsymbol{x}_{t+1}, \sigma_t^2 + y_{t+1}^2, t+1, \boldsymbol{x}_1^T\right) \right)$$

with base case $V\left(\boldsymbol{s}_T, \sigma_T^2, T, \boldsymbol{x}_1^T\right) := -\min_{\theta \in \mathbb{R}^d} \sum_{t=1}^T \left(\theta^\top \boldsymbol{x}_t - y_t\right)^2$. The arguments of $V$ include $\boldsymbol{x}_1^T$ to emphasize the fixed-design setting. Performing the backwards induction generates plays $\hat{y}_1^T$ and $y_1^T$ that witness the value of the game,

$$\min_{\hat{y}_1} \max_{y_1} \cdots \min_{\hat{y}_T} \max_{y_T} \sum_{t=1}^T (\hat{y}_t - y_t)^2 - \min_{\boldsymbol{w} \in \mathbb{R}^d} \sum_{t=1}^T (\boldsymbol{w}^\top \boldsymbol{x}_t - y_t)^2,$$

which is the minimum guaranteeable regret against all data sequences. The resulting minimax strategy is precisely the linear predictor $\hat{y}_{t+1} = \boldsymbol{x}_{t+1}^\top \boldsymbol{P}_{t+1} \boldsymbol{s}_t$, ((MMS)) with coefficient matrices defined by the recursion (1). Note that $\boldsymbol{P}_t$ is a function of every covariate $\boldsymbol{x}_1^T$. The minimax strategy is similar

to follow-the-leader, which would predicts with $\Pi_t^\dagger$ in place of $\boldsymbol{P}_t$; however, $\boldsymbol{P}_t$ is a shrunken version of $\Pi_t^\dagger$ that takes future covariances into account.

The main result of Bartlett et al. [2015] is the minimax optimality of (MMS) for the following classes. For some fixed sequence of positive label budgets $B_1, \ldots, B_T > 0$, define

1. *Label constraints on $y_t$*: $\mathcal{L}(B_1^T) := \{y_1^T : |y_t| \leq B_t \forall \boldsymbol{t} = 1, \ldots, T\}$
2. *Box constraints on $\boldsymbol{x}_t$*: $\mathcal{B}(B_1^T) := \left\{ \boldsymbol{x}_1^T : B_t \geq \sum_{s=1}^{t-1} |\boldsymbol{x}_t^\top \boldsymbol{P}_t \boldsymbol{x}_s| \, B_s \text{ for } 2 \leq t \right\}.$
3. *Ellipsoidal constraints*: $\mathcal{E}(\boldsymbol{x}_1^T, R) := \left\{ y_1^T : \sum_{t=1}^T y_t^2 \boldsymbol{x}_t^\top \boldsymbol{P}_t \boldsymbol{x}_t \leq R \right\}.$

**Theorem 1.** *[Bartlett et al., 2015, Theorems 2 and 10] For each $x_1^T$, the corresponding strategy* (MMS) *is minimax optimal with respect to $\mathcal{B}(B_1^T)$ if $y_1^T \in \mathcal{L}(B_1^T)$ and with respect to $\mathcal{E}(\boldsymbol{x}_1^T, R)$, for any $B_t > 0$ sequence and any $R > 0$, in the following sense:*

(1) *If $\boldsymbol{x}_1^T \in \mathcal{B}(B_1^T)$, then*

$$\sup_{y_1^T \in \mathcal{L}(B_1^T)} R_T((\text{MMS}), \boldsymbol{x}_1^T, y_1^T) = \min_s \sup_{y_1^T \in \mathcal{L}(B_1^T)} R_T(s, \boldsymbol{x}_1^T, y_1^T) = \sum_{t=1}^T B_t^2 \boldsymbol{x}_t^\top \boldsymbol{P}_t \boldsymbol{x}_t,$$

(2)
$$\sup_{y_1^T \in \mathcal{E}(\boldsymbol{x}_1^T, R)} R_T((\text{MMS}), \boldsymbol{x}_1^T, y_1^T) = \min_s \sup_{y_1^T \in \mathcal{E}(\boldsymbol{x}_1^T, R)} R_T(s, \boldsymbol{x}_1^T, y_1^T) = R.$$

## 3 The Forward Algorithm

The previous section described the fixed-design minimax strategy and established sufficient conditions for its optimality. Unfortunately, $\boldsymbol{P}_t$ is recursively defined as a function of the entire $\boldsymbol{x}_1^T$ sequence. In this section, we show that it is possible to remove the fixed-design and known-game-length requirement if we limit the adversary to play sequences that follow the *Adversarial Covariate conditions*. Letting $\mathcal{X}^\infty = \bigcup_{T > 0} (\mathbb{R}^d)^T$ denote the set of covariate sequences of finite length, define

$$\mathcal{A}(\boldsymbol{\Sigma}) := \left\{ \boldsymbol{x}_1^T \in \mathcal{X}^\infty : \text{for } \boldsymbol{P}_0, \ldots, \boldsymbol{P}_T \text{ defined by (1)}, \boldsymbol{P}_0^\dagger \preceq \boldsymbol{\Sigma} \right\}, \text{ and}$$

$$\overline{\mathcal{A}}(\boldsymbol{\Sigma}) := \left\{ \boldsymbol{x}_1^T \in \mathcal{X}^\infty : \text{for } \boldsymbol{P}_0, \ldots, \boldsymbol{P}_T \text{ defined by (1)}, \boldsymbol{P}_0^\dagger = \boldsymbol{\Sigma} \right\}; \tag{2}$$

that is, $\boldsymbol{x}_1^T \in \mathcal{A}(\boldsymbol{\Sigma})$ if the $\boldsymbol{P}_t$ computed by applying (1) to the sequence $\boldsymbol{x}_1^T$ results in $\boldsymbol{P}_0^\dagger \preceq \boldsymbol{\Sigma}$.

The key insight of this section is that it is possible to invert the $\boldsymbol{P}_t$ recursion: we can compute $\boldsymbol{P}_t$ from $\boldsymbol{P}_{t-1}$ and $\boldsymbol{x}_t$. Hence, if we are given $\boldsymbol{P}_0$, then we can compute every $\boldsymbol{P}_t$ online. For some initial condition $\boldsymbol{\Sigma}$, define the *forward recursion* with base case $\boldsymbol{P}_0 = \boldsymbol{\Sigma}^\dagger$ and induction step

$$\boldsymbol{P}_t := \boldsymbol{P}_{t-1} - \frac{a_t}{b_t^2} \boldsymbol{P}_{t-1} \boldsymbol{x}_t \boldsymbol{x}_t^\top \boldsymbol{P}_{t-1}, \text{ where } b_t^2 := \boldsymbol{x}_t^\top \boldsymbol{P}_{t-1} \boldsymbol{x}_t, \quad a_t := \frac{\sqrt{4b_t^2 + 1} - 1}{\sqrt{4b_t^2 + 1} + 1}. \tag{3}$$

The prediction matrix $\boldsymbol{P}_t$ is a function of $\boldsymbol{\Sigma}$ and $\boldsymbol{x}_1^t$ only. For the rest of the paper, we will define (MMS) with respect to the forward recursion, i.e. $\hat{y}_t := \boldsymbol{x}_t^\top \boldsymbol{P}_t \boldsymbol{s}_{t-1}$, where $\boldsymbol{P}_t$ is defined by recursion (3). The calculation of $\hat{y}_t$ only requires knowledge of $\boldsymbol{\Sigma}$, $\boldsymbol{x}_1^t$, and $y_1^{t-1}$, all of which are available to the learner when choosing $\hat{y}_t$. The algorithm needs $O(d^2)$ memory and at each round the computational complexity is $O(d^2)$. It is essential that the two recursions are equivalent, which is guaranteed by the following lemma.

**Lemma 1.** *Let $\boldsymbol{\Sigma} \succeq 0$ be a positive semidefinite matrix. For any covariate sequence $\boldsymbol{x}_1^T \in \overline{\mathcal{A}}(\boldsymbol{\Sigma})$, the $\boldsymbol{P}_t$ matrices defined by the backwards recursion (1) applied to $\boldsymbol{x}_1^T$ are identical to the $\boldsymbol{P}_t$ matrices defined by the forward recursion (3) with base case $\boldsymbol{P}_0 = \boldsymbol{\Sigma}^\dagger$ and updates given by $\boldsymbol{x}_1^T$.*

*Proof.* Let $\boldsymbol{P}_t'$ be defined by the forwards recursion starting from $\boldsymbol{P}_0 = \boldsymbol{\Sigma}^\dagger$ and let $\boldsymbol{P}_t$ be defined by the backwards recursion (1). Our goal is to show that $\boldsymbol{P}_t = \boldsymbol{P}_t'$ for all $t$. The base case $\boldsymbol{P}_0 = \boldsymbol{P}_0'$ is a simple consequence [Bartlett et al., 2015, Lemma 11], which uses repeated applications on Sherman-Morrison to show that

$$\boldsymbol{P}_t^\dagger = \Pi_t + \sum_{s=t+1}^T \frac{\boldsymbol{x}_s^\top \boldsymbol{P}_s \boldsymbol{x}_s}{1 + \boldsymbol{x}_s^\top \boldsymbol{P}_s \boldsymbol{x}_s} \boldsymbol{x}_s \boldsymbol{x}_s^\top. \tag{4}$$

Now, assuming the induction hypothesis $\boldsymbol{P}'_{t-1} = \boldsymbol{P}_{t-1}$, we can evaluate

$$\boldsymbol{P}'_t = \boldsymbol{P}_{t-1} - \frac{a_t}{b_t^2} \boldsymbol{P}_{t-1} \boldsymbol{x}_t \boldsymbol{x}_t^\top \boldsymbol{P}_{t-1}$$

$$= \boldsymbol{P}_t + \boldsymbol{P}_t \boldsymbol{x}_t \boldsymbol{x}_t^\top \boldsymbol{P}_t - \frac{a_t}{b_t^2} \left( \boldsymbol{P}_t + \boldsymbol{P}_t \boldsymbol{x}_t \boldsymbol{x}_t^\top \boldsymbol{P}_t \right) \boldsymbol{x}_t \boldsymbol{x}_t^\top \left( \boldsymbol{P}_t + \boldsymbol{P}_t \boldsymbol{x}_t \boldsymbol{x}_t^\top \boldsymbol{P}_t \right)$$

$$= \boldsymbol{P}_t + \boldsymbol{P}_t \boldsymbol{x}_t \left( 1 - \frac{a_t}{b_t^2} \left( 1 + 2\boldsymbol{x}_t^\top \boldsymbol{P}_t \boldsymbol{x}_t + \left( \boldsymbol{x}_t^\top \boldsymbol{P}_t \boldsymbol{x}_t \right)^2 \right) \right) \boldsymbol{x}_t^\top \boldsymbol{P}_t \qquad (5)$$

By definition, we have $b_t^2 = \boldsymbol{x}_t^\top \boldsymbol{P}_{t-1} \boldsymbol{x}_t = \boldsymbol{x}_t^\top \boldsymbol{P}_t \boldsymbol{x}_t + \left( \boldsymbol{x}_t^\top \boldsymbol{P}_t \boldsymbol{x}_t \right)^2$, which we can invert to find that $\boldsymbol{x}_t^\top \boldsymbol{P}_t \boldsymbol{x}_t = \frac{1}{2} \left( \sqrt{4b_t^2 + 1} - 1 \right)$. Plugging this is, the term in the parenthesis in (5) is

$$1 - \frac{a_t}{b_t^2} \left( 1 + 2\boldsymbol{x}_t^\top \boldsymbol{P}_t \boldsymbol{x}_t + \left( \boldsymbol{x}_t^\top \boldsymbol{P}_t \boldsymbol{x}_t \right)^2 \right) = 1 - \frac{a_t}{b_t^2} \left( 1 + \left( \sqrt{4b_t^2 + 1} - 1 \right) + \frac{1}{4} \left( \sqrt{4b_t^2 + 1} - 1 \right)^2 \right)$$

$$= 1 - \frac{a_t}{b_t^2} \left( \frac{1}{2} \sqrt{4b_t^2 + 1} + \frac{1}{2} + b_t^2 \right)$$

$$= \frac{2}{\sqrt{4b_t^2 + 1} + 1} - \frac{1}{2b_t^2} \left( \sqrt{4b_t^2 + 1} - 1 \right)$$

$$= \frac{4b_t^2 - \left( \sqrt{4b_t^2 + 1} - 1 \right) \left( \sqrt{4b_t^2 + 1} + 1 \right)}{2b_t^2 \left( \sqrt{4b_t^2 + 1} + 1 \right)} = 0,$$

implying that $\boldsymbol{P}'_t = \boldsymbol{P}_t$, as desired. □

Our first result is that this algorithm is actually minimax optimal if we constrain the adversary to play in $\overline{\mathcal{A}}(\boldsymbol{\Sigma})$. Another interpretation is that $\boldsymbol{\Sigma}$ encodes all the necessary scale information the learner needs to respond optimally. That is, (MMS) performs as well as the best strategy that sees the covariate sequence in advance. In particular, knowledge of $\boldsymbol{\Sigma}$, not $T$, is necessary for the learner.

**Theorem 2.** *For all positive semidefinite $\boldsymbol{\Sigma}$, label bounds $B_1, B_2, \ldots > 0$, and constants $b > 0$ and $R > 0$, the minimax strategy (MMS) using the forward recursion (3) starting from $\boldsymbol{P}_0 = \boldsymbol{\Sigma}^\dagger$ is horizon-independent minimax optimal, i.e.*

$$\sup_T \sup_{\boldsymbol{x}_1^T \in \mathcal{X}} \left( \sup_{y_1^T \in \mathcal{Y}(\boldsymbol{x}_1^T)} R_T(s^*, \boldsymbol{x}_1^T, y_1^T) - \min_s \sup_{y_1^T \in \mathcal{Y}(\boldsymbol{x}_1^T)} R_T(s, \boldsymbol{x}_1^T, y_1^T) \right) = 0$$

*for $\left( \mathcal{X}, \mathcal{Y}(\boldsymbol{x}_1^T) \right)$ equal to either $\left( \overline{\mathcal{A}}(\boldsymbol{\Sigma}), \mathcal{E}(\boldsymbol{x}_1^T, R) \right)$ or $\left( \mathcal{B}(B_1^T) \cap \overline{\mathcal{A}}(\boldsymbol{\Sigma}), \mathcal{L}(B_1^T) \right)$.*

*Proof of Theorem 2.* Since $\boldsymbol{x}_1^T \in \overline{\mathcal{A}}(\boldsymbol{\Sigma})$, Lemma 1 implies that the $\boldsymbol{P}_t$ matrices from the forwards and backwards recursions are equivalent, and therefore (MMS) corresponds to the minimax strategy for the fixed-design game with $\boldsymbol{P}_0^\dagger = \boldsymbol{\Sigma}$. The can then apply Theorem 1, part (1), which yields

$$\sup_{y_1^T \in \mathcal{B}(B_1^T)} R_T(s^*, \boldsymbol{x}_1^T, y_1^T) - \min_s \sup_{y_1^T \in \mathcal{B}(B_1^T)} R_T(s, \boldsymbol{x}_1^T, y_1^T) = 0.$$

Since this holds for all $\boldsymbol{x}_1^T$, we actually get the stronger result

$$\sup_T \sup_{\boldsymbol{x}_1^T \in \mathcal{A}(B_1^T) \cap \overline{\mathcal{A}}(\boldsymbol{\Sigma})} \left( \sup_{y_1^T \in \mathcal{B}(B_1^T)} R_T(s^*, \boldsymbol{x}_1^T, y_1^T) - \min_s \sup_{y_1^T \in \mathcal{B}(B_1^T)} R_T(s, \boldsymbol{x}_1^T, y_1^T) \right) = 0.$$

Identical reasoning extends part (2) of Theorem 1 to the adversarial covariate context. □

The adversarial covariate conditions are defined for entire $\boldsymbol{x}_1^T$ sequences, but there is an online characterization, derived from the following lemma.

**Lemma 2.** *Consider any $t \geq 0$, $\boldsymbol{x}_1, \ldots, \boldsymbol{x}_t$, and symmetric matrix $\boldsymbol{P} \succeq 0$. We have that $\boldsymbol{P}^\dagger \succeq \Pi_t$ if and only if, for any $T \geq t + \mathrm{rank}\left( \boldsymbol{P}^\dagger - \Pi_t \right)$, there is a continuation of the covariate sequence, $\boldsymbol{x}_{t+1}, \ldots, \boldsymbol{x}_T$, such that setting $\boldsymbol{P}_t = \boldsymbol{P}$ and defining $\boldsymbol{P}_{t+1}, \ldots, \boldsymbol{P}_T$ by the forward recursion (3) gives $\boldsymbol{P}_T^\dagger = \Pi_T$.*

A stronger version with proof is presented in the Appendix as Theorem 6 and explicitly derives conditions on $\boldsymbol{x}_{t+1}$ that ensure $\boldsymbol{P}^\dagger \succeq \Pi_t$.

In words, a sequence of covariates $\boldsymbol{x}_1^t$ is the prefix of some $\boldsymbol{x}_1^T \in \mathcal{A}(\boldsymbol{\Sigma})$ if $\boldsymbol{P}_s^\dagger \succeq \Pi_s$ for all $s \leq t$, where $\boldsymbol{P}_s$ corresponds to the forward recursion (3) defined by intuition condition $\boldsymbol{P}_0 = \boldsymbol{\Sigma}^\dagger$ and covariates $\boldsymbol{x}_1^t$. Hence, it is equivalent to constrain the adversary to play $\boldsymbol{x}_t$ satisfying this condition at every round, and we do not require the adversary to fix the covariate sequence in advance; it is equivalent to define

$$\mathcal{A}(\boldsymbol{\Sigma}) = \left\{ \boldsymbol{x}_1^T \in \mathcal{X}^\infty : \boldsymbol{P}_0^\dagger = \boldsymbol{\Sigma} \text{ and } \boldsymbol{P}_t^\dagger \succeq \Pi_t \ \forall t \geq 1 \right\}, \text{ and} \tag{6}$$

$$\overline{\mathcal{A}}(\boldsymbol{\Sigma}) = \left\{ \boldsymbol{x}_1^T \in \mathcal{X}^\infty : \boldsymbol{P}_0^\dagger = \boldsymbol{\Sigma}, \boldsymbol{P}_t^\dagger \succeq \Pi_t \ \forall t \geq 1, \text{ and } \boldsymbol{P}_T^\dagger = \Pi_T \right\}. \tag{7}$$

## 4 Expanding the Minimax Conditions

The strategy (MMS) is minimax optimal for any covariate sequence $\boldsymbol{x}_1^T \in \overline{\mathcal{A}}(\boldsymbol{\Sigma})$ if the adversary plays covariates that meet the $\boldsymbol{\Sigma}$ constraint with equality, which is quite restrictive. In this section, we identify a much broader set of constraints on the adversary's actions where (MMS) remains the best learner response. These conditions allow for adversarial design; the data may be chosen in response to the learner's actions.

A natural relaxation is to remove the equality constraints; this results in a set of constraints on the adversary where the labels $\{y_t\}$ are in $\mathcal{L}(\{B_t\}) := \{y_t : |y_t| \leq B_t \forall t \geq 1\}$, and the covariates $\{\boldsymbol{x}_t\}$ are in $\mathcal{A}(\boldsymbol{\Sigma}) \cap \mathcal{B}(\boldsymbol{\Sigma})$, where $\mathcal{B}(\boldsymbol{\Sigma}) = \left\{ \{\boldsymbol{x}_t\} : B_t \geq \sum_{s=1}^{t-1} |\boldsymbol{x}_t^\top \boldsymbol{P}_t \boldsymbol{x}_s| \forall t \geq 1 \right\}$.

The $\mathcal{B}(\boldsymbol{\Sigma})$ condition is necessary for an efficient algorithm [Bartlett et al., 2015], and without the $\mathcal{A}(\boldsymbol{\Sigma})$ condition, the adversary could choose $\boldsymbol{x}_t$ to be a scaled version of $\boldsymbol{s}_{t-1}$ and $y_t = \theta_{t-1}^* \boldsymbol{x}_t$, where $\theta_{t-1}^*$ is the best least squares predictor of $\boldsymbol{x}_1^{t-1}$ and $y_1^{t-1}$. The comparator will never suffer more regret, the algorithm will suffer some regret, and we can scale $x_t$ such that the $\mathcal{B}(\boldsymbol{\Sigma})$ conditions are satisfied. To summarize, without the $\mathcal{A}$ constraint, the adversary can cause arbitrary regret. However, the $\mathcal{A}$ and $\mathcal{B}$ constraints are not sufficient to guarantee a solvable game:

**Lemma 3.** *Fix any $\boldsymbol{\Sigma}$ and any $\{B_t\}$ with $B_t \geq b > 0$ for all $t$. Then, for any $M > 0$, there exists $\boldsymbol{x}_1^T \in \mathcal{A}(\boldsymbol{\Sigma}) \cap \mathcal{B}(\boldsymbol{\Sigma})$ and $y_1^T \in \mathcal{L}(B_1^T)$ such that the minimax regret is larger than $M$.*

A covariate budget is not sufficient for a minimax algorithm; it is not even clear how to define minimax when the regrets are not bounded. Hence, we will introduce *continuation constraints* (the name will become clear soon). Let $\gamma_0 > 0$ be some initial label budget and define $\gamma_t = \gamma_{t-1} - B_t^2 \boldsymbol{x}_t^\top \boldsymbol{P}_t \boldsymbol{x}_t$, with $\boldsymbol{P}_t$ defined by the forward recursion (3). Let $B_\infty(B_1^t) := \{\xi \in \mathbb{R}^t : |\xi_i| \leq B_i, i = 1, \ldots, t\}$ be the hypercube with sides of length $B_1, \ldots, B_t$ and $X_t$ be the matrix with columns $\boldsymbol{x}_1, \ldots, \boldsymbol{x}_t$. For a given covariate budget $\boldsymbol{\Sigma}$ and label budget $\gamma_0$, define the continuation condition

$$\mathcal{C}(\boldsymbol{\Sigma}, \gamma_0) := \left\{ \boldsymbol{x}_1^T : \gamma_t \geq \xi^\top X_t^\top \left( \Pi_t^\dagger - \boldsymbol{P}_t \right) X_t \xi \ \forall \xi \in B_\infty(B_t) \text{ and } t = 1, \ldots, T \right\}, \tag{8}$$

which is equivalent to requiring that $\boldsymbol{s}_t^\top \left( \Pi_t^\dagger - \boldsymbol{P}_t \right) \boldsymbol{s}_t \leq \gamma_t$ for all possible $\boldsymbol{s}_t$.

The rest of this section proves the main result of this paper: if the adversary plays in $\mathcal{ABC}(\boldsymbol{\Sigma}, \gamma_0) := \mathcal{A}(\boldsymbol{\Sigma}) \cap \mathcal{B}(\boldsymbol{\Sigma}) \cap \mathcal{C}(\boldsymbol{\Sigma}, \gamma_0)$, then (MMS) is minimax optimal.

**Theorem 3.** *Consider the two player game defined in Figure 1. For any $\{B_t\} > 0$, $\boldsymbol{\Sigma} \succ 0$ and $\gamma_0 \geq 0$, the player strategy (MMS) has minimax regret $\gamma_0$ and is horizon-independent minimax optimal for $\boldsymbol{x}_1^T \in \mathcal{X} = \mathcal{ABC}(\boldsymbol{\Sigma}, \gamma_0)$ and $y_1^T \in \mathcal{Y} = \mathcal{L}(B_t)$. That is,*

$$\sup_T \left( \sup_{\boldsymbol{x}_1^T \in \mathcal{X}, y_1^T \in \mathcal{Y}} R_T((\text{MMS}), \boldsymbol{x}_1^T, y_1^T) - \min_s \sup_{\boldsymbol{x}_1^T \in \mathcal{X}, y_1^T \in \mathcal{Y}} R_T(s, \boldsymbol{x}_1^T, y_1^T) \right) = 0.$$

We will prove Theorem 3 by first considering adversarial strategies under $\mathcal{A}(\boldsymbol{\Sigma})$ with a fixed game length. We show that, somewhat counterintuitively, the adversary may cause more regret by not using the entire $\boldsymbol{\Sigma}$ budget. Then, we show that the $\mathcal{C}$ condition eliminates these troublesome cases and the adversary exhausts the budget; therefore, the adversary plays $\boldsymbol{x}_1^T \in \overline{\mathcal{A}}(\boldsymbol{\Sigma})$ which implies

that that (MMS) is minimax optimal by results of the previous section. Finally, we note that all the previous arguments apply uniformly across $T$, and since (MMS) is ignorant of $T$, it must be horizon-independent minimax optimal. The $\mathbf{\Sigma}$ constraint, not the game length, seems to be the correct notion of game size.

## 4.1 Limiting $T$

Consider a fixed $T > 0$ and define $\mathcal{A}_T(\mathbf{\Sigma}) := \left\{ \boldsymbol{x}_1^T \in \left(\mathbb{R}^d\right)^T : \boldsymbol{P}_0^\dagger = \mathbf{\Sigma} \text{ and } \boldsymbol{P}_t^\dagger \succeq \Pi_t \ \forall 1 \leq t \leq T \right\}$, the restriction of $\mathcal{A}(\mathbf{\Sigma})$ to sequences of length $T$. This goal of this section is to show i) that it is possible for the adversary to cause more regret by not using up the covariance budget, i.e. $\boldsymbol{P}_T^\dagger \succ \Pi_T$, and ii) that the $\mathcal{C}$ conditions are sufficient to stop this.

We cannot calculate the minimax solution of $\mathcal{A}_T(\mathbf{\Sigma})$ directly. Section G in the appendix explicitly evaluates the first backwards induction step; it is quite complicated and has no closed form solution, and this suggests that efficient backwards induction is unlikely. Instead, we will study the related fixed-design *early-stopping* game. For some fixed $\boldsymbol{x}_1^T$, the game protocol is: at round $t$, the learner predicts $\hat{y}_t$, the adversary chooses $e_t \in \{0, 1\}$ and $y_t \in \mathcal{L}(B_1^T)$. If $e_t = 1$, the learner incurs loss $(\hat{y}_t - y_t)^2$ and the game continues, but if $e_t = 0$, the game ends. Intuitively, the adversary may be able to cause more regret because the learner is regularizing for a covariance budget corresponding to $\boldsymbol{x}_1^T$, and therefore ending the game early causes the learner to over-regularize.

We will derive $\mathcal{C}$ as a condition where the adversary always continues to $T$. In turn, this implies that the adversary will use up the $\mathbf{\Sigma}$ budget in the $\mathcal{A}_T$ game: any $\boldsymbol{x}_1^T$ with remaining $\mathbf{\Sigma}$ budget has a continuation $\boldsymbol{x}_1^{T+k} \in \overline{\mathcal{A}}(\mathbf{\Sigma})$ by Lemma 2, and the $\mathcal{C}$ condition implies that the adversary will continue until $T + k$ and use up the budget. We will make this argument formal.

We begin by defining an incremental version of regret. Define $\Delta_t^* := \min_{\theta \in \mathbb{R}^d} \sum_{s=1}^t (\theta^\top \boldsymbol{x}_s - y_s)^2 - \min_{\theta' \in \mathbb{R}^d} \sum_{s=1}^{t-1} (\theta'^\top \boldsymbol{x}_s - y_s)^2$, the additional loss suffered by the comparator from playing $t$ rounds instead of $t - 1$ rounds. We have $\Delta_t^* \geq 0$ and $L_T^* = \sum_{t=1}^T \Delta_t^*$. The regret of the game with early stopping can be written as $\mathcal{R}_T = \sum_{t=1}^T \left( \prod_{s=1}^t e_s \right) \left( (y_t - \hat{y}_t)^2 - \Delta_t^* \right)$. One might notice that $\delta_t^* = 0$ for the choice $y_t = {\theta_{t-1}^*}^\top \boldsymbol{x}_t$, where $\theta_{t-1}^*$ is the ordinary least squares solution on data through time $t - 1$, and the regret always increases. However, this choice of $y_t$ may violate the label constraints, in particular, for $B_t = 1$ and $x_t \in \mathbb{R}$ increasing. Additionally, we want a constraint where the adversary wants to play *all* remaining rounds, not just the next one, and hence the constraint on $\boldsymbol{y}_t$ will depend on the future covariates.

The value-to-go definition also needs to be adapted to the incremental setting. To this end, we define the instantaneous value-to-go $W(\boldsymbol{s}_t, \sigma_t^2, t, \boldsymbol{x}_1^T)$ by $W(\boldsymbol{s}_T, \sigma_T^2, T, \boldsymbol{x}_1^T) = 0$ and

$$W(\boldsymbol{s}_{t-1}, \sigma_{t-1}^2, t-1, \boldsymbol{x}_1^T) = \max_{e_t \in \{0,1\}} e_t \left( \min_{\hat{y}_t} \max_{y_t} (\hat{y}_t - y_t)^2 - \Delta_t^* + W(\boldsymbol{s}_t, \sigma_t^2, t, \boldsymbol{x}_1^T) \right),$$

where the statistics are updated as $\boldsymbol{s}_t = \boldsymbol{s}_{t-1} + y_t \boldsymbol{x}_t$ and $\sigma_t^2 = \sigma_{t-1}^2 + y_t^2$. It is easy to check that $W_0$ is the minimax regret for this game and that it equals the regret of the fixed design game when the adversary plays every round.

## 4.2 Calculating the Instantaneous Value-to-go

This section derives $\mathcal{C}$ as the condition where $e_t = 1$ for all $t$ and evaluates $W_t$. Throughout, $\mathcal{R}(\boldsymbol{M})$ denotes the row space of matrix $\boldsymbol{M}$. Proofs from this section are heavy on calculation and have been collected in Appendix B. We begin by explicitly calculating $\Delta_t^*$.

**Lemma 4.** *The marginal loss for the comparator of playing another round with covariate $\boldsymbol{x} = \boldsymbol{x}_\parallel + \boldsymbol{x}_\perp$, where $\boldsymbol{x}_\parallel \in \mathcal{R}(\Pi_{t-1})$ and $\boldsymbol{x}_\perp$ is its orthogonal complement, is*

$$\Delta_t^* = y_t^2 \left( 1 - \boldsymbol{x}_t^\top \Pi_t^\dagger \boldsymbol{x}_t \right) - 2 y_t \boldsymbol{s}_{t-1}^\top \Pi_t^\dagger \boldsymbol{x}_t + \left( \boldsymbol{s}_{t-1}^\top \Pi_t^\dagger \boldsymbol{x}_t \right)^2 \frac{\boldsymbol{x}_t^\top \Pi_{t-1}^\dagger \boldsymbol{x}_t}{\boldsymbol{x}_t^\top \Pi_t^\dagger \boldsymbol{x}_t}.$$

**Theorem 4.** *Consider the fixed-design game with early stopping, with covariates $\boldsymbol{x}_1^T$. Define the $\boldsymbol{P}_t$ by the backwards recursion (1) and define $\gamma_t = \sum_{s=t+1}^T B_s^2 \boldsymbol{x}_s^\top \boldsymbol{P}_s \boldsymbol{x}_s$. Suppose that, for all $t$, $\gamma_t \geq \boldsymbol{s}_t^\top \left( \Pi_t^\dagger - \boldsymbol{P}_t \right) \boldsymbol{s}_t$. Then the instantaneous value-to-go is $W(\boldsymbol{s}_t, \sigma_t^2, t, \boldsymbol{x}_1^T) = \boldsymbol{s}_t^\top \left( \boldsymbol{P}_t - \Pi_t^\dagger \right) \boldsymbol{s}_t + \gamma_t$, the adversary causes more regret by continuing the game, and the optimal learner strategy is (MMS).*

*Proof outline.* The proof is by induction, where the base case is easily established with $\gamma_T = 0$ and $\boldsymbol{P}_T = \Pi_T^\dagger$. Now, assuming that $W(\boldsymbol{s}_t, \sigma_t^2, t, \boldsymbol{x}_1^T) = \boldsymbol{s}_t^\top \left( \boldsymbol{P}_t - \Pi_t^\dagger \right) \boldsymbol{s}_t + \gamma_t$, we wish to calculate the $t - 1$ case by evaluating $W(\boldsymbol{s}_{t-1}, \sigma_{t-1}^2, t-1, \boldsymbol{x}_1^T) = \max_{e_t \in \{0,1\}} e_t \left( \min_{\hat{y}_t} \max_{y_t} (\hat{y}_t - y_t)^2 - \Delta_t^* + W_t(\boldsymbol{s}_t, \sigma_t^2, \boldsymbol{x}_1^T) \right)$. We use our expression for $\Delta_t^*$, perform elementary calculations to evaluate the saddle-point, and show that the above evaluates to

$$\max \left\{ \left( \boldsymbol{s}_{t-1}^\top \boldsymbol{P}_t \boldsymbol{x}_t \right)^2 + B_t^2 \boldsymbol{x}_t^\top \boldsymbol{P}_t \boldsymbol{x}_t - \left( \boldsymbol{s}_{t-1}^\top \Pi_t^\dagger \boldsymbol{x}_t \right)^2 \frac{\boldsymbol{x}_t^\top \Pi_{t-1}^\dagger \boldsymbol{x}_t}{\boldsymbol{x}_t^\top \Pi_t^\dagger \boldsymbol{x}_t} + \boldsymbol{s}_{t-1}^\top \left( \boldsymbol{P}_t - \Pi_t^\dagger \right) \boldsymbol{s}_{t-1} + \gamma_t, 0 \right\},$$

which can be shown to always take the first value so long as $\gamma_{t-1} \geq \boldsymbol{s}_{t-1}^\top \left( \Pi_{t-1}^\dagger - \boldsymbol{P}_{t-1} \right) \boldsymbol{s}_{t-1}$. In this case, the induction hypothesis is verified with the $\boldsymbol{P}_t$ update described in the theorem. This implies that the instantaneous value-to-go is always positive and that an optimal adversary will always continue. As a consequence, the covariate sequence $\boldsymbol{x}_1^T \in \overline{\mathcal{A}}(\boldsymbol{P}_0^\dagger)$, which confirms that (MMS) using the forward recursion is minimax optimal via Theorem 2. □

All the ingredients are in place to prove our main result. For convenience, define $\overline{\mathcal{ABC}}(\boldsymbol{\Sigma}, \gamma_0) := \{\boldsymbol{x}_1^T \in \mathcal{ABC}(\boldsymbol{\Sigma}, \gamma_0) : \boldsymbol{P}_T = \Pi_T^\dagger, \gamma_T = 0\}$, the set of sequences that deplete the $\boldsymbol{\Sigma}$ and $\gamma_0$ budgets. Roughly, we will argue that, under $\mathcal{C}(\boldsymbol{\Sigma}, \gamma_0)$, the adversary causes the most regret by playing $\boldsymbol{x}_1^T \in \overline{\mathcal{A}}(\boldsymbol{\Sigma})$, which implies that $\boldsymbol{x}_1^T \in \overline{\mathcal{ABC}}(\boldsymbol{\Sigma}, \gamma_0)$ and the regret is $\gamma_0$. The first step in the analysis is to check that the constraint set is non-trivial.

**Lemma 5.** *Consider the game defined by $\boldsymbol{\Sigma} \succeq 0$, $\gamma_0 \geq \|B_t\|_\infty$ and a $B_t$ sequence. If there exists some $T$ such that $\sum_{t=1}^T \frac{B_t^2}{t + \log(T+1)} \geq \gamma_0$, then there exists a covariate sequence $\boldsymbol{x}_1^T \in \overline{\mathcal{ABC}}(\boldsymbol{\Sigma}, \gamma_0)$. In particular, any $B_t$ that are bounded below satisfy this condition.*

In reasoning about optimal strategies, Theorem 4 allows us to easily establish conditions when the learner is playing suboptimally and could be causing more regret. However, Theorem 4 applies to a fixed design game that is allowed to stop early, and we wish to reason about the adversarial covariate case. The next lemma makes the crucial connection.

**Lemma 6.** *Suppose $\boldsymbol{x}_1^t \in \mathcal{ABC}(\boldsymbol{\Sigma}, \gamma_0)$ but $\gamma_t > 0$. Then there exists an extension $\boldsymbol{x}_{t+1}, \ldots, \boldsymbol{x}_T$ in $\mathcal{ABC}(\boldsymbol{\Sigma}, \gamma_0)$ with $\boldsymbol{x}_1^T \in \overline{\mathcal{A}}(\boldsymbol{\Sigma}, \gamma_0)$ and $W(\boldsymbol{s}_t, \sigma_t^2, t, \boldsymbol{x}_1^T) = \boldsymbol{s}_t^\top (\boldsymbol{P}_t - \Pi_t^\dagger) \boldsymbol{s}_t + \gamma_t$ equal to the instantaneous value-to-go.*

The proof is a simple consequence of checking that the extension Lemma 2 is compatible with condition $\mathcal{C}$. We can now prove the minimax optimality of (MMS) on the $\mathcal{ABC}$ game.

*Proof of Theorem 3.* We will show something stronger: the optimal adversary strategy for the game in Figure 1 plays an $\boldsymbol{x}_1^T$ sequence in $\overline{\mathcal{ABC}}$ and causes exactly $\gamma_0$ regret against (MMS).

First, assume that the game stops before round $T + 1$ and $\boldsymbol{x}_1, \ldots, \boldsymbol{x}_T$ have been played. There are four possible scenarios depending on whether the $\boldsymbol{\Sigma}$ or $\gamma_0$ budgets are exhausted.

**Case: both budgets exhausted.** In this case, $\boldsymbol{x}_1^T \in \overline{\mathcal{ABC}}(\boldsymbol{\Sigma}, \gamma_0)$ and optimal holds by results from Section 3.

**Case: neither budget exhausted.** We apply Lemma 2 to conclude that there exists a covariate sequence $\boldsymbol{x}_{T+1}^{T+k}$ that uses up the $\boldsymbol{\Sigma}$ budget. The $\mathcal{C}(\boldsymbol{\Sigma}, \gamma_0)$ constraint guarantees that the adversary can cause more regret by playing these rounds. Hence, an adversary that exhausts neither budget is suboptimal.

**Case: only $\boldsymbol{\Sigma}$ budget exhausted.** Since $\boldsymbol{P}_t - \Pi_t^\dagger \succeq 0$, we cannot exhaust the $\gamma_0$ before the $\boldsymbol{\Sigma}$ budget and still satisfy the $\mathcal{C}$ constraint.

**Case: only $\gamma_0$ budget exhausted.** If the $\boldsymbol{\Sigma}$ budget is exhausted, then $\boldsymbol{x}_1^T \in \mathcal{A}$ and hence the minimax regret is $\sum_{t=1}^T B_t^2 \boldsymbol{x}_t^\top \boldsymbol{P}_t \boldsymbol{x}_t$ by Theorem 2. Since $\gamma_T = \gamma_0 - \sum_{t=1}^T B_t^2 \boldsymbol{x}_t^\top \boldsymbol{P}_t \boldsymbol{x}_t$, the adversary strategy is suboptimal if $\gamma_T > 0$ since it is possible to cause $\gamma_0$ regret. These arguments cover all four cases, we can conclude that the adversary can cause at most $\gamma_0$ regret and that any strategy that causes $\gamma_0$ regret must exhaust the $\boldsymbol{\Sigma}$ and $\gamma_0$ budgets.

In all cases, the adversary can cause at most $\gamma_0$ regret and it is necessary for the adversary to play $\boldsymbol{x}_1^T \in \overline{\mathcal{ABC}}(\boldsymbol{\Sigma}, \gamma_0)$, which implies that (MMS) is optimal. In other words, for $\boldsymbol{x}_1^T \in \mathcal{X} = \mathcal{ABC}(\boldsymbol{\Sigma}, \gamma_0)$ and $y_1^T \in \mathcal{Y} = \mathcal{L}(B_t)$, we have

$$\sup_{\boldsymbol{x}_1^T \in \mathcal{X}, y_1^T \in \mathcal{Y}} R_T((\text{MMS}), \boldsymbol{x}_1^T, y_1^T) - \min_s \sup_{\boldsymbol{x}_1^T \in \mathcal{X}, y_1^T \in \mathcal{Y}} R_T(s, \boldsymbol{x}_1^T, y_1^T) = 0$$

for all $T > 0$, which implies the result. □

**The Necessity of a $\gamma_0$ Bound** Requiring a $\gamma_0$ bound may seem artificial at first, especially since it translates directly into a bound on the regret. However, it is a reasonable constraint to impose, for several reasons. First, recall that Lemma 3 argues that the regret of just the $\mathcal{A}(\boldsymbol{\Sigma}) \cap \mathcal{B}(\boldsymbol{\Sigma})$ game is infinite. Second, the restriction on the adversary is mild: if $\boldsymbol{x}_1^T \in \mathcal{ABC}(\boldsymbol{\Sigma}, \gamma_0)$, then $\boldsymbol{x}_1^T \in \mathcal{ABC}(\boldsymbol{\Sigma}, \gamma')$ for $\gamma' \geq \gamma_0$, and so the budget can be adjusted online. Finally, we emphasize that the learner does not need to know $\gamma_0$ to play (MMS).

## 5 Follow the Regularized Leader

The minimax strategy (MMS) can be interpreted as playing follow-the-regularized-leader with a certain data-dependent regularizer.

**Lemma 7.** *The minimax strategy* (MMS) *is exactly follow-the-regularized-leader, predicting* $\hat{y}_t = \theta^\top \boldsymbol{x}_t$ *at round* $t$, *where regularization matrices* $\boldsymbol{R}_t$ *are*

$$\boldsymbol{R}_0 := \boldsymbol{P}_0^{-1}, \text{ and } \boldsymbol{R}_t := \boldsymbol{R}_{t-1} + \frac{1}{1 + \boldsymbol{x}_t^\top \boldsymbol{P}_t \boldsymbol{x}_t} \boldsymbol{x}_t \boldsymbol{x}_t^\top - \boldsymbol{x}_{t-1} \boldsymbol{x}_{t-1}^\top, \tag{9}$$

*and* $\theta$ *is the solution to* $\min_\theta \sum_{s=1}^{t-1} (\theta^\top \boldsymbol{x}_s - y_s)^2 + \theta^\top \boldsymbol{R}_t \theta$.

It is also possible to derive a $\boldsymbol{R}_t$ recursion without referring to $\boldsymbol{P}_t$; see Lemma 11. For comparison, the last step minimax algorithm [Azoury and Warmuth, 2001] plays $\hat{y}_t = \left(\sum_{s=1}^{t} \boldsymbol{x}_s \boldsymbol{x}_s^\top\right)^{-1} \boldsymbol{s}_{t-1}$, so we can also view the minimax algorithm as last step minimax with a regularization of $\sum_{s=t+1}^{T} \frac{\boldsymbol{x}_s^\top \boldsymbol{P}_s \boldsymbol{x}_s}{1 + \boldsymbol{x}_s^\top \boldsymbol{P}_s \boldsymbol{x}_s} \boldsymbol{x}_s \boldsymbol{x}_s^\top$.

We have shown that for the adversarial covariates protocol with $\mathcal{X} = \mathcal{ABC}(\boldsymbol{\Sigma}, \gamma_0)$, (MMS) is the minimax optimal strategy and receives $\gamma_0$ regret. Our last result helps quantify this regret by proving a $O(\log(T))$ regret bound for the games analyzed in Section 3.

**Theorem 5.** *For any fixed* $T$ *and* $B_1^T$, *the minimax regret of the box-constrained game has the bound*

$$\sup_{\boldsymbol{x}_1^T \in \overline{\mathcal{A}}(\boldsymbol{\Sigma})} \sup_{y_1^T \in \mathcal{L}(B_1^T)} R_T(s^*, \boldsymbol{x}_1^T, y_1^T) \leq \frac{d\|B_1^T\|_\infty}{\|\boldsymbol{\Sigma}\|_2} \left(1 + 2\ln\left(1 + \frac{\|\boldsymbol{\Sigma}\|_2^2}{2\|B_1^T\|_\infty^2}\|B_1^T\|_2^2\right)\right).$$

## 6 Conclusion

We have presented the minimax optimal strategy for online linear regression where the covariate and label sequence are chosen adversarially and the measure of game length is a covariance budget instead of the number of rounds. Because the strategy has access to a more informative measure of game size, $\boldsymbol{\Sigma}$, it can compete with strategies that know the number of rounds. The minimax strategy is efficient and only needs to update $\boldsymbol{P}_t$ and $\boldsymbol{s}_t$.

One could interpret the results of our paper as finding a more natural way to measure the length of the game that admits a tractable minimax strategy. What other game protocols can be reparameterized to admit efficient minimax strategies? As a general method, one could start with minimax algorithms for constrained cases then search for parameterizations which preserve the optimality.

We have also provided an intuitive view of the algorithm as follow-the-regularized-leader with a specific data-dependent regularizer. This interpretation can be used to bound the excess regret when the budget $\boldsymbol{\Sigma}$ is misspecified, perhaps allowing for adaptation to $\boldsymbol{\Sigma}$.

## Acknowledgements

We gratefully acknowledge the support of the NSF through grant IIS-1619362.

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
