[Supplementary Material · appendix.pdf]

## A  Extra Lemmas

**Lemma 8.** *Let $x_1^T$ be any covariate sequence and $P_1, \ldots, P_T$ the associated precision matrices given by the backwards recursion* (1). *For any invertible matrix $W \in \mathbb{R}^{d \times d}$, let $x_t' = W x_t$. Then the precision matrices of $x_1', \ldots, x_T'$ are exactly $P_t' = W^{\dagger^\top} P_t W^\dagger$ and $x_t^\top P_t x_t = x_t'^\top P_t' x_t'$.*

*Proof.* First, we can easily check that $P_T' = \left( \sum_{t=1}^T x_t' x_t'^\top \right)^\dagger = (W^\top)^\dagger \left( \sum_{t=1}^T x_t x_t^\top \right)^\dagger W^\dagger$. Now, assume that the hypothesis holds for $t$. Then

$$
\begin{aligned}
P_{t-1}' &= P_t' + \tilde{P}_t x_t' x_t'^\top P_t' \\
&= W^{\dagger^\top} P_t W^\dagger + W^{\dagger^\top} \left( P_t W^\dagger W x_t x_t W^\top W^{\dagger^\top} P_t \right) W^\dagger \\
&= W^{\dagger^\top} P_{t-1} W^\dagger.
\end{aligned}
$$

$\square$

## B  Calculating $\Delta_t^*$

While the update of $P_t$ is given by the forward recursion, the rank one update of $\Pi_t$ is more complicated; Sherman-Morrison cannot be used directly.

**Lemma 9.** *Using $x_\perp := x - \Pi_{t-1} \Pi_{t-1}^\dagger x$ to denote the projection of $x_t$ onto the orthogonal complement of $\Pi_{t-1}$, we have*

$$
\Pi_t^\dagger = \begin{cases}
\Pi_{t-1}^\dagger - \dfrac{x_\perp x^\top \Pi_{t-1}^\dagger + \Pi_{t-1}^\dagger x x_\perp^\top}{x_\perp^\top x_\perp} + \dfrac{x_\perp \left( 1 + x^\top \Pi_{t-1}^\dagger x \right) x_\perp^\top}{(x_\perp^\top x_\perp)^2} & \text{if } x \notin \mathcal{C}(\Pi_{t-1}), \text{ and} \\[3ex]
\Pi_{t-1}^\dagger + \dfrac{\Pi_{t-1}^\dagger x_t x_t^\top \Pi_{t-1}^\dagger}{1 - x_t^\top \Pi_{t-1}^\top x_t^\top} & \text{otherwise .}
\end{cases}
$$

*Proof.* We will write $X$ as the matrix with columns $x_1, \ldots, x_{t-1}$. Thus, we have

$$
\Pi_t = \Pi_{t-1} + x x^\top = \begin{bmatrix} X & x \end{bmatrix} \begin{bmatrix} X^\top \\ x^\top \end{bmatrix},
$$

and since $X$ has linearly independent columns, (without loss of generality; we shall see why later), $\begin{bmatrix} X & x \end{bmatrix}$ has linearly independent columns since $x$ is not in the column space of $X$. Therefore, we have

$$
\begin{bmatrix} X & x \end{bmatrix}^\dagger = \left( \begin{bmatrix} X^\top \\ x^\top \end{bmatrix} \begin{bmatrix} X & x \end{bmatrix} \right)^{-1} \begin{bmatrix} X^\top \\ x^\top \end{bmatrix}
$$

and

$$
\Pi_t^\dagger = \left( \Pi_{t-1} + x x^\top \right)^\dagger = \begin{bmatrix} X & x \end{bmatrix} \left( \begin{bmatrix} X^\top \\ x^\top \end{bmatrix} \begin{bmatrix} X & x \end{bmatrix} \right)^{-2} \begin{bmatrix} X^\top \\ x^\top \end{bmatrix}.
$$

Now, recall that the matrix that projects onto the column space of $X$ is $\mathcal{P} := X X^\dagger$ and define $x_\| := \mathcal{P} x$ and $x_\perp = x - x_\|$. We can calculate the middle matrix by using the block matrix inversion formula:

$$
\begin{aligned}
\left( \begin{bmatrix} X^\top \\ x^\top \end{bmatrix} \begin{bmatrix} X & x \end{bmatrix} \right)^{-1} &= \begin{bmatrix} (X^\top X)^{-1} + \frac{X^\dagger x x^\top X^{\top\dagger}}{x^\top x - x^\top \mathcal{P} x} & \frac{-X^\dagger x}{x^\top x - x^\top \mathcal{P} x} \\ \frac{-x^\top X^{\top\dagger}}{x^\top x - x^\top \mathcal{P} x} & \frac{1}{x^\top x - x^\top \mathcal{P} x} \end{bmatrix} \\
&= \frac{1}{x^\top x - x_\|^\top x_\|} \begin{bmatrix} (X^\top X)^{-1} \left( x^\top x - x_\|^\top x_\| \right) + X^\dagger x x^\top X^{\top\dagger} & -X^\dagger x \\ -x^\top X^{\top\dagger} & 1 \end{bmatrix},
\end{aligned}
$$

and so

$$\left(\begin{bmatrix}\boldsymbol{X}^\top\\\boldsymbol{x}^\top\end{bmatrix}\begin{bmatrix}\boldsymbol{X} & \boldsymbol{x}\end{bmatrix}\right)^{-1}\begin{bmatrix}\boldsymbol{X}^\top\\\boldsymbol{x}^\top\end{bmatrix} = \frac{1}{\boldsymbol{x}^\top\boldsymbol{x}-\boldsymbol{x}_\|^\top\boldsymbol{x}_\|}\begin{bmatrix}\boldsymbol{X}^\dagger\left(\boldsymbol{x}^\top\boldsymbol{x}-\boldsymbol{x}_\|^\top\boldsymbol{x}_\|\right)-\boldsymbol{X}^\dagger\boldsymbol{x}\boldsymbol{x}_\perp^\top\\\boldsymbol{x}_\perp^\top\end{bmatrix}.$$

Using the Pythagorean theorem (i.e. that $\boldsymbol{x}^\top\boldsymbol{x} = \boldsymbol{x}_\|^\top\boldsymbol{x}_\| + \boldsymbol{x}_\perp^\top\boldsymbol{x}_\perp$) and that $\Pi_{t-1}^\dagger = \boldsymbol{X}^{\top\dagger}\boldsymbol{X}^\dagger$, we have

$$
\begin{aligned}
\Pi_t^\dagger &= \frac{1}{\left(\boldsymbol{x}_\perp^\top\boldsymbol{x}_\perp\right)^2}\begin{bmatrix}\boldsymbol{X}^{\dagger\top}\boldsymbol{x}_\perp^\top\boldsymbol{x}_\perp-\boldsymbol{x}_\perp\boldsymbol{x}^\top\boldsymbol{X}^{\top\dagger} & \boldsymbol{x}_\perp\end{bmatrix}\begin{bmatrix}\boldsymbol{X}^\dagger\boldsymbol{x}_\perp^\top\boldsymbol{x}_\perp-\boldsymbol{X}^\dagger\boldsymbol{x}\boldsymbol{x}_\perp^\top\\\boldsymbol{x}_\perp^\top\end{bmatrix}\\
&= \frac{1}{\left(\boldsymbol{x}_\perp^\top\boldsymbol{x}_\perp\right)^2}\left(\Pi_{t-1}^\dagger\left(\boldsymbol{x}_\perp^\top\boldsymbol{x}_\perp\right)^2-\boldsymbol{x}_\perp^\top\boldsymbol{x}_\perp\left(\boldsymbol{x}_\perp\boldsymbol{x}^\top\Pi_{t-1}^\dagger+\Pi_{t-1}^\dagger\boldsymbol{x}\boldsymbol{x}_\perp^\top\right)\right)\\
&\quad + \frac{\boldsymbol{x}_\perp\boldsymbol{x}^\top\Pi_{t-1}^\dagger\boldsymbol{x}\boldsymbol{x}_\perp^\top}{\left(\boldsymbol{x}_\perp^\top\boldsymbol{x}_\perp\right)^2}+\frac{\boldsymbol{x}_\perp\boldsymbol{x}_\perp^\top}{\left(\boldsymbol{x}_\perp^\top\boldsymbol{x}_\perp\right)^2}\\
&= \Pi_{t-1}^\dagger-\frac{\boldsymbol{x}_\perp\boldsymbol{x}^\top\Pi_{t-1}^\dagger+\Pi_{t-1}^\dagger\boldsymbol{x}\boldsymbol{x}_\perp^\top}{\boldsymbol{x}_\perp^\top\boldsymbol{x}_\perp}+\frac{\boldsymbol{x}_\perp\left(1+\boldsymbol{x}^\top\Pi_{t-1}^\dagger\boldsymbol{x}\right)\boldsymbol{x}_\perp^\top}{(\boldsymbol{x}_\perp^\top\boldsymbol{x}_\perp)^2}.
\end{aligned}
$$

Thus, we can evaluate

$$
\begin{aligned}
\boldsymbol{x}^\top\Pi_t^\dagger\boldsymbol{x} &= \boldsymbol{x}^\top\Pi_{t-1}^\dagger\boldsymbol{x}-\frac{\boldsymbol{x}^\top\boldsymbol{x}_\perp\boldsymbol{x}^\top\Pi_{t-1}^\dagger\boldsymbol{x}+\boldsymbol{x}^\top\Pi_{t-1}^\dagger\boldsymbol{x}\boldsymbol{x}_\perp^\top\boldsymbol{x}}{\boldsymbol{x}_\perp^\top\boldsymbol{x}_\perp}+\frac{\boldsymbol{x}^\top\boldsymbol{x}_\perp\left(1+\boldsymbol{x}^\top\Pi_{t-1}^\dagger\boldsymbol{x}\right)\boldsymbol{x}_\perp^\top\boldsymbol{x}}{(\boldsymbol{x}_\perp^\top\boldsymbol{x}_\perp)^2}\\
&= \boldsymbol{x}^\top\Pi_{t-1}^\dagger\boldsymbol{x}-2\boldsymbol{x}^\top\Pi_{t-1}^\dagger\boldsymbol{x}+1+\boldsymbol{x}^\top\Pi_{t-1}^\dagger\boldsymbol{x}\\
&= 1,
\end{aligned}
$$

and

$$
\begin{aligned}
\boldsymbol{x}^\top\Pi_t^\dagger\boldsymbol{s} &= \boldsymbol{x}^\top\Pi_{t-1}^\dagger\boldsymbol{s}-\boldsymbol{x}^\top\Pi_{t-1}^\dagger\boldsymbol{s}-\frac{\boldsymbol{x}^\top\Pi_{t-1}^\dagger\boldsymbol{x}\boldsymbol{x}_\perp^\top\boldsymbol{s}}{\boldsymbol{x}_\perp^\top\boldsymbol{x}_\perp}+\frac{\left(1+\boldsymbol{x}^\top\Pi_{t-1}^\dagger\boldsymbol{x}\right)\boldsymbol{x}_\perp^\top\boldsymbol{s}}{\boldsymbol{x}_\perp^\top\boldsymbol{x}_\perp}\\
&= -\frac{\boldsymbol{x}^\top\Pi_{t-1}^\dagger\boldsymbol{x}\boldsymbol{x}_\perp^\top\boldsymbol{s}}{\boldsymbol{x}_\perp^\top\boldsymbol{x}_\perp}+\frac{\left(1+\boldsymbol{x}^\top\Pi_{t-1}^\dagger\boldsymbol{x}\right)\boldsymbol{x}_\perp^\top\boldsymbol{s}}{\boldsymbol{x}_\perp^\top\boldsymbol{x}_\perp}\\
&= 0.
\end{aligned}
$$

Finally, notice that

$$\boldsymbol{s}^\top\Pi_t^\dagger\boldsymbol{s} = \boldsymbol{s}^\top\Pi_{t-1}^\dagger\boldsymbol{s}$$

since $\boldsymbol{x}_\perp^\top\boldsymbol{s} = 0$.

The second case is a consequence of the Sherman-Morrison formula. Since $\Pi_t$, $\Pi_{t-1}$, and $\boldsymbol{x}_t$ are all in the same eigenspace, we can without loss of generality assume full rank and apply Sherman-Morrison. A precise formulation can also be found in e.g. Harville [1997]. $\square$

**Lemma 10.** *For a PSD symmetric matrix* $\Pi$, $s \in \mathcal{R}(\Pi)$ *and* $x \notin \mathcal{R}(\Pi)$*, we have*

$$
\begin{aligned}
x^\top\left(\Pi+xx^\top\right)^\dagger x &= 1,\\
s^\top\left(\Pi+xx^\top\right)^\dagger x &= 0,\\
s^\top\left(\Pi+xx^\top\right)^\dagger s &= s^\top\Pi^\dagger s.
\end{aligned}
$$

*Proof.* Write the small SVD $\Pi = U\Lambda U^\top$ (that is, diagonal $\Lambda$, $U$ with orthonormal columns). Choose a unit vector $v$, vectors $a$ and $b$, and scalar $\alpha \neq 0$ so that

$$U^\top v = 0, \qquad\qquad x = Ua+\alpha v, \qquad\qquad s = Ub.$$

Define

$$w = \left(\Pi + xx^\top\right)^\dagger x, \qquad\qquad r = \left(\Pi + xx^\top\right)^\dagger s.$$

Because these are the minimal norm solutions to the linear equations

$$\left(\Pi + xx^\top\right) w = x, \qquad\qquad \left(\Pi + xx^\top\right) r = s,$$

we can certainly write

$$w = Uc + \beta v, \qquad\qquad r = Ud + \gamma v,$$

for some vectors $c$ and $d$ and scalars $\beta$ and $\gamma$. Then we have

$$\left(U\Lambda U^\top + (Ua + \alpha v)(Ua + \alpha v)^\top\right)(Uc + \beta v) = Ua + \alpha v$$
$$\Leftrightarrow \qquad U\Lambda c + (Ua + \alpha v)(a^\top c + \alpha\beta) = Ua + \alpha v$$
$$\Leftrightarrow \qquad \Lambda c + a(a^\top c + \alpha\beta) = a, \qquad \alpha(a^\top c + \alpha\beta) = \alpha,$$
$$\Leftrightarrow \qquad c = 0, \qquad \beta = 1/\alpha.$$

Similarly,

$$\left(U\Lambda U^\top + (Ua + \alpha v)(Ua + \alpha v)^\top\right)(Ud + \gamma v) = Ub$$
$$\Leftrightarrow \qquad U\Lambda d + (Ua + \alpha v)(a^\top d + \alpha\gamma) = Ub$$
$$\Leftrightarrow \qquad \Lambda d + a(a^\top d + \alpha\gamma) = b, \qquad \alpha(a^\top d + \alpha\gamma) = 0,$$
$$\Leftrightarrow \qquad d = \Lambda^{-1}b, \qquad \gamma = -a^\top\Lambda^{-1}b/\alpha$$

Thus,

$$x^\top\left(\Pi + xx^\top\right)^\dagger x = x^\top w = (Ua + \alpha v)^\top (1/\alpha)v = 1.$$
$$s^\top\left(\Pi + xx^\top\right)^\dagger x = s^\top w = (Ub)^\top (1/\alpha)v = 0.$$
$$s^\top\left(\Pi + xx^\top\right)^\dagger s = s^\top r = (Ub)^\top\left(U\Lambda^{-1}b - a^\top\Lambda^{-1}b/\alpha v\right) = b^\top\Lambda^{-1}b = s^\top U\Lambda^{-1}U^\top s = s^\top\Pi^\dagger s.$$

$$\square$$

We can also verify these calculations directly using Lemma 9, but more intuition can be gleaned from the proof above.

## C  Missing Proofs

*Proof of Lemma 3.* It suffices to consider a one dimensional game. Fix some $T$ and consider the simplest data sequence $x_t = b$. Applying the alternative form of $P_t$ from (4), we can check that $\sum_{s=1}^t x_t^\top P_t x_s \le x_t^\top \Pi_t^{-1} \sum_{s=1}^t x_s \le b$, so the $\mathcal{B}(\{B_t\})$ conditions hold. Theorem 1 implies that the minimax regret is exactly $\sum_{t=1}^T B_t^2 x_t^\top P_t x_t \ge b^2 \sum_{t=1}^T x_t^\top P_t x_t$.

We can explicitly write out the recursion for $x_t^\top P_t x_t$ in this simple case. The initial value is $x_T^2 P_T = x_T^2/(\sum_{t=1}^T x_t^2) = T^{-1}$, and the recursion becomes $x_{t-1}^2 P_{t-1} = x_t^2 P_t + (x_t^2 P_t)^2$. Denoting $z_t = x_t^2 P_t$, we see that $z_T = T^{-1}$ and $z_{t-1} = z_t + z_t^2$. This exact recursion was analyzed in [Takimoto and Warmuth, 2000, Lemma 3], which proved that $\sum_{t=1}^T z_t \ge \log(T) - \log\log(T)$, which implies that, for the data sequence $x_1, \ldots, x_T = 1$, $\mathcal{R} \ge b^2(\log(T) - \log\log(T))$. Hence, for a given $M$, we can always find a $T$ large enough to make $\mathcal{R} > M$.

The last step is to check that the $\mathcal{A}(\Sigma)$ condition is satisfied. We exploit the fact that $x_t^\top P_t x_t$ is scale invariant; it does not change when $x_t$ is multiplied by any invertible matrix, as the $P_t$ term will be pre and post-multiplied by the inverse of this matrix. This result appeared in [Bartlett et al., 2015], but we include a proof in Lemma 8 in the Appendix for completeness.

The scale invariance implies that the data sequence $x_t' = cx_t$, for any $c$, has the same regret as if the adversary played $x_t$ with the same labels. Hence, if the $P_0$ of $x_t$ violates the $\Sigma$ condition, then we may choose $c$ large enough such that $c^{-2}P_0$, which is the $P_0$ corresponding to $x_t'$, does not. Since the regret remains the same, and the $\mathcal{B}$ conditions are also scale invariance, our $x_t'$ sequence verifies the claim of the lemma. $\square$

*Proof of Lemma 4.* We have

$$\Delta_t^* = \sigma_t^2 - \sigma_{t-1}^2 - (s_{t-1} + y_t x_t)^\top \Pi_t^\dagger (s_{t-1} + y_t x_t) + s_{t-1}^\top \Pi_{t-1}^\dagger s_{t-1}$$
$$= y_t^2 - 2y_t s_{t-1} \Pi_t^\dagger x_t - y_t^2 x_t^\top \Pi_t^\dagger x_t + s_{t-1}^\top \left( \Pi_{t-1}^\dagger - \Pi_t^\dagger \right) s_{t-1}.$$

First, assume that $x_\perp = 0$. Then $x_t$ is in the column space of $\Pi_t$ and $\Pi_{t-1}$, and an application of the generalized Sherman-Morrison formula (see e.g. Harville [1997]) yields

$$\Pi_{t-1}^\dagger = \left( \Pi_t - x_t x_t^\top \right)^\dagger = \Pi_t^\dagger + \frac{\Pi_t^\dagger x_t x_t^\top \Pi_t^\dagger}{1 - x_t^\top \Pi_t^\dagger x_t}, \tag{10}$$

and so

$$\Delta_t^* = y_t^2 \left( 1 - x_t^\top \Pi_t^\dagger x_t \right) - 2y_t s_{t-1}^\top \Pi_t^\dagger x_t + \frac{\left( s_{t-1}^\top \Pi_t^\dagger x_t \right)^2}{1 - x_T^\top \Pi_t^\dagger x_t}.$$

Finally, notice that (10) implies

$$x_t^\top \Pi_{t-1}^\dagger x_t = \frac{x_t^\top \Pi_t^\dagger x_t}{1 - x_t^\top \Pi_t^\dagger x_t}.$$

when $x_\perp = 0$, yielding the claim in that case.

Now, assume that $x_\perp \neq 0$. Then

$$(s_{t-1} + y_t x_t)^\top \Pi_t^\dagger (s_{t-1} + y_t x_t)$$
$$= s_{t-1}^\top \Pi_t^\dagger s_{t-1} + 2y_t s_{t-1}^\top \Pi_t^\dagger x_t + y_t^2 x_t^\top \Pi_t^\dagger x_t$$
$$= s_{t-1}^\top \Pi_{t-1}^\dagger s_{t-1} + y_t^2,$$

where we applied the three claims of Lemma 10 to obtain the second equality. Therefore, $\Delta_t^* = 0$, and our formula is correct. $\qquad\square$

*Proof of Theorem 4.* The proof is by induction: assume that $W(s_t, \sigma_t^2, t, \Pi_t) = s_t^\top \left( P_t - \Pi_t^\dagger \right) s_t + \gamma_t$. The base case is easily established with $\gamma_T = 0$ and $P_T = \Pi_T^\dagger$ yielding the base case of $W(\cdot, \cdot, 0, \cdot) = 0$. Now, we assume $W$ is correct at round $t$ and want to verify the formula at $t-1$. Hence, under the usual definitions of $s_t$ and $\sigma_t^2$, we can calculate

$$W(s_{t-1}, \sigma_{t-1}^2, t-1, x_1^T)$$
$$= \max_{e_t \in \{0,1\}} e_t \left( \min_{\hat{y}_t} \max_{y_t} (\hat{y}_t - y_t)^2 - \Delta_t^* + W(s_t, \sigma_t^2, t, x_1^T) \right)$$
$$= \left( \min_{\hat{y}} \max_{y} (\hat{y} - y)^2 - y^2 \left( 1 - x_t^\top \Pi_t^\dagger x_t \right) + 2y s_{t-1}^\top \Pi_t^\dagger x_t - \left( s_{t-1}^\top \Pi_t^\dagger x_t \right)^2 \frac{x_t^\top \Pi_{t-1}^\dagger x_t}{x_t^\top \Pi_t^\dagger x_t} \right.$$
$$\left. + (s_{t-1} + y x_t)^\top \left( P_t - \Pi_t^\dagger \right) (s_{t-1} + y x_t) + \gamma_t \right)_+$$
$$= \left( \min_{\hat{y}} \max_{y} \hat{y}^2 + 2y \left( s_{t-1}^\top \Pi_t^\dagger x_t + s_{t-1}^\top \left( P_t + \Pi_t^\dagger \right) x_t - \hat{y} \right) + y^2 x_t^\top \Pi_t^\dagger x_t \right.$$
$$\left. - \left( s_{t-1}^\top \Pi_t^\dagger x_t \right)^2 \frac{x_t^\top \Pi_{t-1}^\dagger x_t}{x_t^\top \Pi_t^\dagger x_t} + s_{t-1}^\top \left( P_t - \Pi_t^\dagger \right) s_{t-1} + y^2 x_t^\top \left( P_t - \Pi_t^\dagger \right) x_t + \gamma_t \right)_+$$
$$= \left( \min_{\hat{y}} \max_{y} \hat{y}^2 + 2y \left( s_{t-1}^\top P_t x_t - \hat{y} \right) + y^2 x_t^\top P_t x_t \right.$$
$$\left. - \left( s_{t-1}^\top \Pi_t^\dagger x_t \right)^2 \frac{x_t^\top \Pi_{t-1}^\dagger x_t}{x_t^\top \Pi_t^\dagger x_t} + s_{t-1}^\top (P_t - \Pi_t^\dagger) s_{t-1} + \gamma_t \right)_+.$$

The objective is convex in $y$ and therefore the optimum will be on the boundary at $\pm B_t$. Thus,

$$W(s_{t-1}, \sigma_{t-1}^2, t-1, \boldsymbol{x}_1^T) = \left( \min_{\hat{y}} \hat{y}^2 + 2B_t \left| \boldsymbol{s}_{t-1}^\top \boldsymbol{P}_t \boldsymbol{x}_t - \hat{y} \right| - B_t^2 \boldsymbol{x}_t^\top \boldsymbol{A}_t \boldsymbol{x}_t \right.$$

$$\left. - \left( \boldsymbol{s}_{t-1}^\top \Pi_t^\dagger \boldsymbol{x}_t \right)^2 \frac{\boldsymbol{x}_t^\top \Pi_{t-1}^\dagger \boldsymbol{x}_t}{\boldsymbol{x}_t^\top \Pi_t^\dagger \boldsymbol{x}_t} + \boldsymbol{s}_{t-1}^\top (\boldsymbol{P}_t - \Pi_t^\dagger) \boldsymbol{s}_{t-1} + \gamma_t \right)_+ .$$

This objective is convex in $\hat{y}$ as well, and hence we can minimize it by setting the subgradient to zero. Under the condition that $\left| \boldsymbol{s}_{t-1}^\top \boldsymbol{B}_t \boldsymbol{x}_t \right| \leq B_t$, the subgradient at $\hat{y} = \boldsymbol{s}_{t-1}^\top \boldsymbol{P}_t \boldsymbol{x}_t$ contains zero, so

$$W(s_{t-1}, \sigma_{t-1}^2, t-1, \boldsymbol{x}_1^T)$$

$$= \left( \left( \boldsymbol{s}_{t-1}^\top \boldsymbol{P}_t \boldsymbol{x}_t \right)^2 + B_t^2 \boldsymbol{x}_t^\top \boldsymbol{P}_t \boldsymbol{x}_t - \left( \boldsymbol{s}_{t-1}^\top \Pi_t^\dagger \boldsymbol{x}_t \right)^2 \frac{\boldsymbol{x}_t^\top \Pi_{t-1}^\dagger \boldsymbol{x}_t}{\boldsymbol{x}_t^\top \Pi_t^\dagger \boldsymbol{x}_t} + \boldsymbol{s}_{t-1}^\top \left( \boldsymbol{P}_t - \Pi_t^\dagger \right) \boldsymbol{s}_{t-1} + \gamma_t \right)_+ .$$

If $\boldsymbol{x}_t \in \mathcal{R}(\Pi_{t-1})$, then we can use a generalized Sherman-Morrison lemma (see Lemma 9 for details) to calculate $\boldsymbol{x}_t^\top \Pi_{t-1}^\dagger \boldsymbol{x}_t = \frac{\boldsymbol{x}_t^\top \Pi_t^\dagger \boldsymbol{x}_t}{1 - \boldsymbol{x}_t^\top \Pi_t^\dagger \boldsymbol{x}_t}$, and therefore

$$\left( \boldsymbol{s}_{t-1}^\top \Pi_t^\dagger \boldsymbol{x}_t \right)^2 \frac{\boldsymbol{x}_t^\top \Pi_{t-1}^\dagger \boldsymbol{x}_t}{\boldsymbol{x}_t^\top \Pi_t^\dagger \boldsymbol{x}_t} + \boldsymbol{s}_{t-1}^\top \Pi_t^\dagger \boldsymbol{s}_{t-1} = \boldsymbol{s}_{t-1}^\top \left( \Pi_t^\dagger \boldsymbol{x}_t \boldsymbol{x}_t^\top \Pi_t^\dagger \frac{1}{1 - \boldsymbol{x}_t^\top \Pi_t^\dagger \boldsymbol{x}_t} + \Pi_t^\dagger \right) \boldsymbol{s}_{t-1}$$

$$= \boldsymbol{s}_{t-1} \Pi_{t-1}^\dagger \boldsymbol{s}_{t-1}.$$

If instead $\boldsymbol{x}_t \notin \mathcal{R}(\Pi_{t-1})$, then a standard fact for the ordinary least squares solution is $\boldsymbol{s}_{t-1}^\top \Pi_t^\dagger \boldsymbol{x}_t = 0$ and $\boldsymbol{s}_{t-1}^\top \Pi_t^\dagger \boldsymbol{s}_{t-1} = \boldsymbol{s}_{t-1}^\top \Pi_{t-1}^\dagger \boldsymbol{s}_{t-1}$ (a proof of this fact is provided in Lemma 10). In either case, we have

$$W(s_{t-1}, \sigma_{t-1}^2, t-1, \boldsymbol{x}_1^T) = \left( \boldsymbol{s}_{t-1}^\top \left( \boldsymbol{P}_t + \boldsymbol{P}_t \boldsymbol{x}_t \boldsymbol{x}_t^\top \boldsymbol{P}_t \right) \boldsymbol{s}_{t-1} + B_t^2 \boldsymbol{x}_t^\top \boldsymbol{P}_t \boldsymbol{x}_t - \boldsymbol{s}_{t-1}^\top \Pi_{t-1}^\dagger \boldsymbol{s}_{t-1} + \gamma_t \right)_+$$

$$= \left( \boldsymbol{s}_{t-1}^\top \left( \boldsymbol{P}_{t-1} - \Pi_{t-1}^\dagger \right) \boldsymbol{s}_{t-1} + \gamma_{t-1} \right)_+ ,$$

verifying the $\boldsymbol{P}_t$ and $\gamma_t$ recurrence. If $\gamma_{t-1} \geq \boldsymbol{s}_{t-1}^\top \left( \Pi_{t-1}^\dagger - \boldsymbol{P}_{t-1} \right) \boldsymbol{s}_{t-1}$ holds for all $t$, then the instantaneous value-to-go is always positive, an optimal adversary will always continue, and the data sequence seen by the learner is $\boldsymbol{x}_1^T \in \overline{\mathcal{A}}(\boldsymbol{P}_0)$. In this case, the minimax strategy is confirmed to be (MMS) by Theorem 2. $\qquad \square$

*Proof of Lemma 5.* It actually suffices to take the simplest of sequences, $\boldsymbol{x}_t = \boldsymbol{e}_1$. For any fixed $T$, $\boldsymbol{P}_T = \frac{1}{T} \boldsymbol{e}_1 \boldsymbol{e}_1^\top$, where all the $\boldsymbol{P}_t$ for the remainder of the proof are with respect to the covariate sequence of $T$ copies of $\boldsymbol{e}_1$. In this case, the $\boldsymbol{P}_t$ matrices are all zero except for the first element which evolves like $\boldsymbol{P}_{t-1} = \boldsymbol{P}_t + \boldsymbol{P}_t^2$. This is the same recursion studied by Takimoto and Warmuth [2000], who proved a lower bound of $(t + \log(T + 1) - \log(t + 1))^{-1}$. Thus, we can bound

$$\sum_{t=1}^T B_t^2 \boldsymbol{x}_t^\top \boldsymbol{P}_t \boldsymbol{x}_t \geq \sum_{t=1}^T \frac{B_t^2}{t + \log(T + 1) - \log(t + 1)} \geq \sum_{t=1}^T \frac{B_t^2}{t + \log(T + 1)},$$

and thus the assumption that $\sum_{t=1}^T \frac{B_t^2}{t + \log(T+1)} \geq \gamma_0$ implies that there is an $\boldsymbol{x}_1^T$ sequence that produces an upper bound on $\gamma_0$.

Next, notice that if we choose any index $t'$ with $B_{t'} \leq \|B_t\|_\infty$, then the covariate sequence $\boldsymbol{x}_t = \boldsymbol{e}_1 \{t = t'\}$, where $\{\cdot\}$ is the indicator function, produces $\sum_{t=1}^T B_t^2 \boldsymbol{x}_t^\top \boldsymbol{P}_t \boldsymbol{x}_t = B_{t'}^2 \leq \gamma_0$. Now, $\sum_{t=1}^T B_t^2 \boldsymbol{x}_t^\top \boldsymbol{P}_t \boldsymbol{x}_t \leq \gamma_0$ is a continuous function of $\boldsymbol{x}_1^T$, and hence, by the intermediate value theorem, there is a $\boldsymbol{x}_1^T$ with $\sum_{t=1}^T B_t^2 \boldsymbol{x}_t^\top \boldsymbol{P}_t \boldsymbol{x}_t = \gamma_0$.

Next, we check the $\mathcal{B}$ constraint. First, observe that it suffices to check that we can construct some $\boldsymbol{x}_1^T$ using the construction of the previous paragraph. On $[0, 1/2]$, $x/(1 + x) \geq x/2$ and the $\boldsymbol{P}_t$ sequence is decreasing, so $\sum_{s=t+1}^T \boldsymbol{x}_s^2 \frac{\boldsymbol{x}_s^2 \boldsymbol{P}_s}{1 + \boldsymbol{x}_s^2 \boldsymbol{P}_s} \geq \frac{1}{2} x \sum_{s=t+1}^T \boldsymbol{x}_s^4 \boldsymbol{P}_s$, and combined with (4), we have

$$\sum_{s=1}^t \left| \boldsymbol{x}_t^\top \boldsymbol{P}_s \boldsymbol{x}_s \right| \leq |\boldsymbol{x}_t| \frac{\sum_{s=1}^t |\boldsymbol{x}_s|}{\Pi_t + \sum_{s=t+1}^T \boldsymbol{x}_s^2 \frac{\boldsymbol{x}_s^2 \boldsymbol{P}_s}{1 + \boldsymbol{x}_s^2 \boldsymbol{P}_s}} \leq |\boldsymbol{x}_t| \frac{\sum_{s=1}^t |\boldsymbol{x}_s|}{\Pi_t + \sum_{s=t+1}^T \boldsymbol{x}_s^2 \frac{\boldsymbol{x}_s^2 \boldsymbol{P}_s}{2}}.$$

The arguments from the previous section show that $\sum_{s=t+1}^{T} x_s^2 \frac{x_s^2 P_s}{2}$ can be made to grow without bound (in particular, by taking $\boldsymbol{x}_s = \boldsymbol{e}_1$), and so we can always find a long enough covariate sequence such that the $\mathcal{B}$ constraint is met.

Now, fix any $\boldsymbol{x}_1^T$ sequence that achieves the $\mathcal{B}$ and $\mathcal{C}$ constraints. By Lemma 8, we can, for any invertible matrix $\boldsymbol{A}$, rescale the covariate sequence to form $\boldsymbol{x}_t' = \boldsymbol{A}\boldsymbol{x}_t$ to obtain the corresponding $\boldsymbol{P}_t' = \boldsymbol{W}^{-1} \boldsymbol{P}_t \boldsymbol{W}^{-1}$. Since we have $\boldsymbol{x}_s^\top \boldsymbol{P}_t \boldsymbol{x}_t = \boldsymbol{x}_s'^\top \boldsymbol{P}_t' \boldsymbol{x}_t'$ for any $s$ and $t$, the $\mathcal{B}$ and $\mathcal{C}$ constraints hold automatically. Therefore, we are free to choose $\boldsymbol{A}$ such that $\boldsymbol{P}_0' = \boldsymbol{\Sigma}$, and therefore $\boldsymbol{x}_1^{T'} \in \mathcal{ABC}(\boldsymbol{\Sigma}, \gamma_0)$.

$\square$

*Proof of Lemma 7.* Since $\theta$ minimizes a convex unconstrained objective, we set the derivative to zero and obtain the solution $\theta^* = \left( \sum_{s=1}^{t-1} \boldsymbol{x}_s \boldsymbol{x}_s^\top + \boldsymbol{R}_t \right)^{-1} \boldsymbol{s}_{t-1}$. Thus, we need to verify that $\sum_{s=1}^{t-1} \boldsymbol{x}_s \boldsymbol{x}_s^\top + \boldsymbol{R}_t = \boldsymbol{P}_t^{-1}$ for all $t$. This also guarantees that $\boldsymbol{R}_t \succeq 0$. The $t = 0$ case is by definition of $\hat{R}_0$. Now, proceeding by induction, assume that the statement holds for $t - 1$. Then,

$$\sum_{s=1}^{t-1} \boldsymbol{x}_s \boldsymbol{x}_s^\top + \boldsymbol{R}_t = \sum_{s=1}^{t-1} \boldsymbol{x}_s \boldsymbol{x}_s^\top + \boldsymbol{R}_{t-1} + \frac{\boldsymbol{x}_t \boldsymbol{x}_t^\top}{1 + \boldsymbol{x}_t^\top \boldsymbol{P}_t \boldsymbol{x}_t} - \boldsymbol{x}_{t-1} \boldsymbol{x}_{t-1}^\top$$

$$= \sum_{s=1}^{t-2} \boldsymbol{x}_s \boldsymbol{x}_s^\top + \boldsymbol{R}_{t-1} + \frac{\boldsymbol{x}_t \boldsymbol{x}_t^\top}{1 + \boldsymbol{x}_t^\top \boldsymbol{P}_t \boldsymbol{x}_t} = \boldsymbol{P}_{t-1}^{-1} + \frac{\boldsymbol{x}_t \boldsymbol{x}_t^\top}{1 + \boldsymbol{x}_t^\top \boldsymbol{P}_t \boldsymbol{x}_t} = \boldsymbol{P}_t^{-1},$$

where the last equality is by Sherman-Morrison. $\square$

## D  Auxiliary Lemmas and Theorems

**Lemma 2** For any $t \geq 0$, $\boldsymbol{x}_1, \ldots, \boldsymbol{x}_t$, and symmetric matrix $\boldsymbol{P} \succeq 0$, the following two conditions are equivalent:

1. $\boldsymbol{P}^\dagger \succeq \Pi_t$
2. For any $T \geq t + k$, where $k = \operatorname{rank}\left( \boldsymbol{P}^\dagger - \Pi_t \right)$, there is a continuation of the covariate sequence, $\boldsymbol{x}_{t+1}, \ldots, \boldsymbol{x}_T$, such that setting $\boldsymbol{P}_t = \boldsymbol{P}$ and defining $\boldsymbol{P}_{t+1}, \ldots, \boldsymbol{P}_T$ by the forward recursion (3) gives $\boldsymbol{P}_T^\dagger = \Pi_T$.

*Proof.* To see that Condition 1 implies Condition 2, we will consider the forward algorithm recursion, starting from $\boldsymbol{P}_t = \boldsymbol{P}$, and show that we can find suitable covariate vectors $\boldsymbol{x}_{t+1}, \ldots, \boldsymbol{x}_{t+k}$, so that

$$\operatorname{rank}\left( \boldsymbol{P}_{t+i}^\dagger - \sum_{s=1}^{t+i} \boldsymbol{x}_s \boldsymbol{x}_s^\top \right) = k - i,$$

which implies the result for $T = t + k$. It suffices to show that, at each step, we can reduce this rank by one. Consider the spectral decomposition

$$\boldsymbol{P}^\dagger - \Pi_t = \sum_{i=1}^{m} \lambda_i \boldsymbol{v}_i \boldsymbol{v}_i^\top,$$

for orthonormal $v_1, \ldots, v_k$ and non-negative $\lambda_1 \geq \cdots \geq \lambda_k > 0$. Choosing $\boldsymbol{x}_{t+1} = \beta \boldsymbol{v}_k$, there is a $\beta \geq 0$ such that

$$\boldsymbol{P}_{t+1}^\dagger - \Pi_{t-1} = \sum_{i=1}^{k-1} \lambda_i \boldsymbol{v}_i \boldsymbol{v}_i^\top,$$

which implies the result. Indeed, we have

$$\boldsymbol{P}_{t+1}^\dagger - \Pi_{t+1} = \boldsymbol{P}_t^\dagger + \frac{a_{t+1} \beta^2}{(1 - a_{t+1}) b_{t+1}^2} \boldsymbol{v}_k \boldsymbol{v}_k^\top - \Pi_t - \beta^2 \boldsymbol{v}_{t+1} \boldsymbol{v}_{t+1}^\top$$

$$= \sum_{i=1}^{k-1} \lambda_i \boldsymbol{v}_i \boldsymbol{v}_i^\top + \left( \lambda_k - \beta^2 + \frac{a_{t+1} \beta^2}{(1 - a_{t+1}) b_{t+1}^2} \right) \boldsymbol{v}_k \boldsymbol{v}_k^\top.$$

Recall

$$b_{t+1}^2 = \boldsymbol{x}_{t+1}^\top \boldsymbol{P}_t \boldsymbol{x}_{t+1}$$

$$= \beta^2 \boldsymbol{v}_k^\top \left( \Pi_t + \sum_{i=1}^k \lambda_i \boldsymbol{v}_i \boldsymbol{v}_i^\top \right)^\dagger \boldsymbol{v}_k$$

$$= \beta^2 c^2,$$

where we have defined $c^2 > 0$. We need to choose $\beta \geq 0$ so that

$$\lambda_k = \beta^2 \left( 1 - \frac{a_{t+1}}{(1 - a_{t+1}) b_{t+1}^2} \right)$$

$$= \beta^2 \left( 1 - \frac{\sqrt{4 b_t^2 + 1} - 1}{2 b_t^2} \right)$$

$$= \beta^2 \left( 1 - \frac{\sqrt{4 \beta^2 c^2 + 1} - 1}{2 \beta^2 c^2} \right)$$

$$\Leftrightarrow \qquad c^2 \lambda_k = \beta^2 c^2 - \frac{\sqrt{4 \beta^2 c^2 + 1} - 1}{2}.$$

Since $c^2 \lambda_k \geq 0$ and the function on the right hand side maps to $[0, \infty)$ for $\beta \geq 0$, there is a suitable choice of $\beta$. To see that this implies the result for any $T \geq t + k$, notice that by choosing a smaller value of $\beta$, the rank is not diminished.

To see the other direction, notice that Condition 2 and Lemma 1 together imply that there is a $T$ and a completion of the sequence, $\boldsymbol{x}_1, \ldots, \boldsymbol{x}_T$, so that plugging the sequence into the backwards recurrence (1) gives $\boldsymbol{P}_t = \boldsymbol{P}$. But then Equation (4) shows that

$$\boldsymbol{P}_t^\dagger = \Pi_t + \sum_{s=t+1}^T \frac{\boldsymbol{x}_s^\top \boldsymbol{P}_s \boldsymbol{x}_s}{1 + \boldsymbol{x}_s^\top \boldsymbol{P}_s \boldsymbol{x}_s} \boldsymbol{x}_s \boldsymbol{x}_s^\top \succeq \Pi_t,$$

which is Condition 1. □

**Lemma 11.** *The definition of $\boldsymbol{R}_t$ in Equation (9) is equivalent to defining $\boldsymbol{R}_0 = \boldsymbol{P}_0^{-1}$ and*

$$\boldsymbol{R}_t = \boldsymbol{R}_{t-1} + \frac{2 \boldsymbol{x}_t \boldsymbol{x}_t^\top}{\sqrt{1 + 4 \boldsymbol{x}_t^\top \left( \boldsymbol{R}_{t-1} + \sum_{s=1}^{t-2} \boldsymbol{x}_s \boldsymbol{x}_s^\top \right)^{-1} \boldsymbol{x}_t} + 1} - \boldsymbol{x}_{t-1} \boldsymbol{x}_{t-1}^\top. \qquad (11)$$

*Proof.* First, we can calculate

$$4 b_t^2 = \left( \frac{2}{1 - a_t} - 1 \right)^2 - 1 = \left( \frac{1 + a_t}{1 - a_t} \right)^2 - 1 = \frac{4 a_t}{(1 - a_t)^2}, \qquad (12)$$

which implies that $b_t^2 = \frac{a_t}{(1 - a_t)^2}$. Using the forward recursion 1 of $\boldsymbol{P}_t$, we have

$$\boldsymbol{x}_t^\top \boldsymbol{P}_t \boldsymbol{x}_t = b_t^2 - a_t b_t^2 = \frac{a_t}{1 - a_t},$$

and

$$\frac{1}{1 + \boldsymbol{x}_t^\top \boldsymbol{P}_t \boldsymbol{x}_t} = 1 - a_t = \frac{2}{\sqrt{1 + 4 b_t^2} + 1},$$

which, when combined $b_t^2 = \boldsymbol{x}_t^\top \left( \boldsymbol{R}_{t-1} + \sum_{s=1}^{t-2} \boldsymbol{x}_s \boldsymbol{x}_s^\top \right)^{-1} \boldsymbol{x}_t$, yields the desired statement. □

# E  The Proof of the Regret Bound

This section proves Theorem 5, which is quoted below for convenience.

**Theorem 5**  For any fixed $T$ and $B_1^T$, we can bound the minimax regret of the box-constrained game by

$$\sup_{\boldsymbol{x}_1^T \in \overline{\mathcal{A}}(\boldsymbol{\Sigma})} \sup_{y_1^T \in \mathcal{L}(B_1^T)} R_T(s^*, \boldsymbol{x}_1^T, y_1^T) \leq \frac{d\|B_1^T\|_\infty}{\|\boldsymbol{\Sigma}\|_2}\left(1 + 2\ln\left(1 + \frac{\|\boldsymbol{\Sigma}\|_2^2}{2\|B_1^T\|_\infty^2}\|B_1^T\|_2^2\right)\right).$$

The minimax analysis shows that the minimax regret is equal to $\sup_{\boldsymbol{x}_1^T \in \mathcal{A}(\boldsymbol{\Sigma})} \sum_t B_t^2 \boldsymbol{x}_t^\top \boldsymbol{P}_t \boldsymbol{x}_t$, which we bound by defining the worst case regret function,

$$\phi_t(\boldsymbol{\Sigma}, B_1^t) = \max_{\boldsymbol{x}_1,\ldots,\boldsymbol{x}_t}\left\{\sum_{s=1}^{t} B_s^2 \boldsymbol{x}_s^\top \boldsymbol{P}_s(\boldsymbol{x}_1,\ldots,\boldsymbol{x}_s)\boldsymbol{x}_s : \boldsymbol{\Sigma} \succeq \boldsymbol{P}_t(\boldsymbol{x}_1,\ldots,\boldsymbol{x}_t)\sum_{s=1}^{t}\boldsymbol{x}_s\boldsymbol{x}_s^\top\right\}.$$

We drop the explicit dependence of $\boldsymbol{P}_t$ on $\boldsymbol{x}_1^T$ and reparameterize by $r_t^2 = \boldsymbol{P}_t\boldsymbol{x}_t\boldsymbol{x}_t^\top$:

$$\phi_t(\boldsymbol{\Sigma}, B_1^t) = \max_{\boldsymbol{r}_1,\ldots,\boldsymbol{r}_t}\left\{\sum_{s=1}^{t} B_s^2 \operatorname{tr}(r_t^2) : \boldsymbol{\Sigma} \succeq \boldsymbol{P}_t\sum_{s=1}^{t}\boldsymbol{P}_s^{-1}r_s^2\right\}.$$

We then relax the optimization to allow $\boldsymbol{r}_t$ to be a general matrix and argue that the worst case regret function is upper bounded by $d$ 1-dimensional functions.

Noting that $\boldsymbol{P}_{t-1}\boldsymbol{P}_t^{-1} = \boldsymbol{I} + \boldsymbol{P}_t\boldsymbol{x}_t\boldsymbol{x}_t^\top = \boldsymbol{I} + r_t^2$, we can derive an induction for $\phi_t$:

$$\phi_t(\boldsymbol{\Sigma}, B_1^t) = \max_{\boldsymbol{x}_1,\ldots,\boldsymbol{x}_t} B_t^2\boldsymbol{x}_t^\top\boldsymbol{P}_t\boldsymbol{x}_t + \left\{\sum_{s=1}^{t-1}B_s^2\boldsymbol{x}_s^\top\boldsymbol{P}_s\boldsymbol{x}_s : \boldsymbol{\Sigma} - \boldsymbol{P}_t\boldsymbol{x}_t\boldsymbol{x}_t^\top \succeq \boldsymbol{P}_t\sum_{s=1}^{t-1}\boldsymbol{x}_s\boldsymbol{x}_s^\top\right\}$$

$$= \max_{\boldsymbol{x}_1,\ldots,\boldsymbol{x}_t} B_t^2\boldsymbol{x}_t^\top\boldsymbol{P}_t\boldsymbol{x}_t + \left\{\sum_{s=1}^{t-1}B_s^2\boldsymbol{x}_s^\top\boldsymbol{P}_s\boldsymbol{x}_s : (\boldsymbol{\Sigma} - \boldsymbol{P}_t\boldsymbol{x}_t\boldsymbol{x}_t^\top)\boldsymbol{P}_{t-1}\boldsymbol{P}_t^{-1} \succeq \boldsymbol{P}_{t-1}\sum_{s=1}^{t-1}\boldsymbol{x}_s\boldsymbol{x}_s^\top\right\}$$

$$= \max_{\boldsymbol{r}_t,\ldots,\boldsymbol{r}_t} B_t^2\operatorname{tr}(r_t^2) + \left\{\sum_{s=1}^{t-1}B_s^2\operatorname{tr}(r_s^2) : (\boldsymbol{\Sigma} - r_t^2)(\boldsymbol{I} + r_s^2) \succeq \boldsymbol{P}_{t-1}\sum_{s=1}^{t-1}\boldsymbol{x}_s\boldsymbol{x}_s^\top\right\}$$

$$= \max_{\boldsymbol{r}_t} B_t^2\operatorname{tr}(r_t^2) + \phi_{t-1}\left((\boldsymbol{\Sigma} - r_t^2)(\boldsymbol{I} + r_s^2), B_1^{t-1}\right).$$

As a first step, we will bound $\phi_t$ in one dimension where $\phi_t(\Sigma, B_1^t) = \max_{r_t} B_t^2 r_t^2 + \phi_{t-1}((\Sigma - r_t^2)(1 + r_t^2), B_1^{t-1})$. We have omitted the bolding to emphasize that we are in the scalar case. The following lemma borrows heavily from [Bartlett et al., 2015, Theorem 5]; the proof is in

**Lemma 12.** *For every $T$ and every $B_1^T$ with $\|B_1^T\|_\infty \leq \Sigma$,*

$$\phi_T(\Sigma, B_1^T) \leq \min\left\{-\ln(1 - \Sigma), 1 + 2\log\left(1 + \frac{\|B_1^T\|_2^2}{2}\right)\right\}.$$

*Proof.* In fact, we will prove the slightly stronger statement: for any positive function $f(T)$ with $f(0) \geq 0$ and $B_{T+1}^2 e^{-f(T)/2} + f(T) \leq f(T+1)$, we have

$$\phi_T(\Sigma, B_1^T) \leq \min\{-\ln(1 - \Sigma), f(T)\}.$$

We prove this by induction on $T$. The base case is trivial. Assume that the induction hypothesis holds for $T$. Then,

$$\phi_{T+1}(\Sigma, B_1^T) = \max_{r_{T+1}^2} B_{T+1}^2 r_{T+1}^2 + \phi_T\left((\Sigma - r_t^2)(1 + r_t^2), B_1^T\right)$$

$$= \max_{0 \leq x \leq \Sigma} B_{T+1}^2 \frac{\sqrt{(1 + \Sigma)^2 - 4x} - (1 - \Sigma)}{2} + \phi_T(x, B_1^{T-1})$$

$$\leq \max_{0 \leq x \leq \Sigma} B_{T+1}^2 \frac{\sqrt{(1 + \Sigma)^2 - 4x} - (1 - \Sigma)}{2} + \min\{-\ln(1 - x), f(T)\}.$$

Define $\hat{x} = 1 - \exp(-f(T))$, which is where the minimum switches from the first to the second argument. To find the maximizing $x$, we will calculate when the derivative is positive:

$$\frac{-B_{T+1}^2}{\sqrt{(1+\Sigma)^2 - 4x}} + \frac{1}{1-x} \geq 0$$
$$\Leftrightarrow (1+\Sigma)^2 - 4x - B_{T+1}^4(1-x)^2 \geq 0$$
$$\Leftrightarrow (1+\Sigma)^2 - B_{T+1}^4(1+x)^2 + 4(B_{T+1}^4 - 1)x \geq 0, \tag{13}$$

which is true for all $x \leq \Sigma$ and $B^4 \leq \Sigma$. In fact, $B_{T+1}^4$ may be bigger than $\Sigma$ without violating the constraint, but in particular $B_t \leq \Sigma$ is enough.

The sign of the derivative changes at $\hat{x}$. If $\Sigma \leq \hat{x}$, then the maximum is at $\Sigma$ and we have

$$\phi_{T+1}(\Sigma, B_1^T) \leq B_{T+1}^2 \frac{\sqrt{(1+\Sigma)^2 - 4\Sigma} - (1-\Sigma)}{2} + \phi_T(\Sigma)$$
$$= \phi_T(\Sigma).$$

Otherwise, if $\hat{x} \leq \Sigma$, the maximum is at $\hat{x}$ and we have

$$\phi_{T+1}(\Sigma, B_1^T) \leq B_{T+1}^2 \frac{\sqrt{(1+\Sigma)^2 - 4\hat{x}} - (1-\Sigma)}{2} + f(T)$$
$$\leq B_{T+1}^2 \sqrt{1 - \hat{x}} + f(T)$$
$$= B_{T+1}^2 \exp(-f(T)/2) + f(T)$$

where the second line was from using $\Sigma \leq 1$. This allows any $f(T)$ that satisfies

$$B_{T+1}^2 e^{-f(T)/2} + f(T) \leq f(T+1).$$

To check that $f(T) = 1 + 2\log(1 + 1/2\sum_t B_t^2)$ indeed works, we calculate:

$$f(T+1) - f(T) = -2\log\left(\frac{2 + \sum_{t=1}^{T} B_t^2}{2 + \sum_{t=1}^{T+1} B_t^2}\right)$$
$$= -2\log\left(1 - \frac{B_{T+1}^2}{2 + \sum_{t=1}^{T+1} B_t^2}\right)$$
$$\geq \frac{B_{T+1}^2}{1 + \frac{1}{2}\sum_{t=1}^{T+1} B_t^2}$$
$$\geq e^{-1/2}\frac{B_{T+1}^2}{1 + \frac{1}{2}\sum_{t=1}^{T+1} B_t^2}$$
$$= B_{T+1}^2 e^{-f(T)/2}.$$

$\square$

The general multidimensional case can be bounded by first relaying the assumption that $r_t^2 = \boldsymbol{P}_t \boldsymbol{x}_t \boldsymbol{x}_t^\top$ to allow general matrices $\boldsymbol{R}_t$, which only increases the value of the maximization. We can then apply the one-dimensional bound in every direction:

**Lemma 13.** *For any $\Sigma \geq 0$, $\psi_t(\Sigma \boldsymbol{I}, B_1^t) = \sum_{i=1}^d \phi_t(\Sigma, B_1^t)$, where $\phi_t(\Sigma)$ is the one-dimensional regret bound.*

*Proof.* The base case is trivial since both sides are zero. For the inductive hypothesis, assume that $\psi_{t-1}(\Sigma \boldsymbol{I}, B_1^{t-1}) = \sum_{i=1}^d \phi_{t-1}(\Sigma, B_1^{t-1})$. Denoting the eigenvalues of $\boldsymbol{R}$ by $\lambda_1, \ldots, \lambda_d$, we have

$$\psi_t(\Sigma \boldsymbol{I}, B_1^t) = \max_{\boldsymbol{R}} B_t^2 \operatorname{tr}(\boldsymbol{R}) + \psi_{t-1}\left((\Sigma \boldsymbol{I} - \boldsymbol{R})(\boldsymbol{I} + \boldsymbol{R}), B_1^{t-1}\right)$$
$$= \max_{\boldsymbol{R}}\left\{\sum_{i=1}^d \lambda_i + \sum_{i=1}^d \phi_{t-1}\left((1+\lambda_i)(\Sigma - \lambda_i), B_1^{t-1}\right)\right\} = \sum_{i=1}^d \phi_t(\Sigma, B_1^t).$$

$\square$

*Proof of Theorem 5.* Recall from Theorem 1 that for given $T$ and $\boldsymbol{x}_1, \ldots, \boldsymbol{x}_T$, the regret of the box constrained game is precisely $\sum_{t=1}^{T} B_t^2 \boldsymbol{x}_t^\top \boldsymbol{P}_t \boldsymbol{x}_t$. Lemma 13 bounds $\sum_{t=1}^{T} B_t^2 \boldsymbol{x}_t^\top \boldsymbol{P}_t \boldsymbol{x}_t$ by a quantity that does not depend on $\boldsymbol{x}_t$. To invoke Lemma 12, we need that $B_t \leq \max_i \lambda_i$ for all $t$, which is exactly $||B_1^T||_\infty \leq ||\boldsymbol{\Sigma}||_2$. Rescaling the $B_t$ sequence (and hence the regret bound) gives the result. $\qquad \square$

# F  Explicit Constraints on $x_t$

We have seen that the learner is minimax as long as the adversary plays a covariate sequence that is in $\mathcal{A}(\boldsymbol{\Sigma})$. The following theorem provides explicit constraints on the choice of $\boldsymbol{x}_{t+1}$ as a function of the past covariates.

**Theorem 6.** *The consistency condition*

$$\boldsymbol{P}_{t+1}^{-1} - \sum_{q=1}^{t+1} \boldsymbol{x}_q \boldsymbol{x}_q^\top \succeq 0$$

*is equivalent to the conjunction of*

1. $\boldsymbol{P}_t^{-1} - \sum_{q=1}^{t} \boldsymbol{x}_q \boldsymbol{x}_q^\top \succeq 0$,

2. $\boldsymbol{x}_{t+1}$ *is orthogonal to the kernel of* $\boldsymbol{P}_t^{-1} - \sum_{q=1}^{t} \boldsymbol{x}_q \boldsymbol{x}_q^\top$, *and*

3. $\boldsymbol{x}_{t+1}^\top \boldsymbol{P}_t \boldsymbol{x}_{t+1} \leq d_t(\hat{\boldsymbol{x}}_{t+1}) + \sqrt{d_t(\hat{\boldsymbol{x}}_{t+1})}$,

*where* $\hat{\boldsymbol{x}}_{t+1} = \boldsymbol{x}_{t+1}/||\boldsymbol{x}_{t+1}||$ *and*

$$d_t(\hat{\boldsymbol{x}}) = \frac{\hat{\boldsymbol{x}}^\top \boldsymbol{P}_t \hat{\boldsymbol{x}}}{\hat{\boldsymbol{x}}^\top \left( \boldsymbol{P}_t^{-1} - \sum_{q=1}^{t} \boldsymbol{x}_q \boldsymbol{x}_q^\top \right)^\dagger \hat{\boldsymbol{x}}}.$$

Notice that $0 \leq d_t(\hat{\boldsymbol{x}}) \leq 1$.

*Proof.* The $\boldsymbol{x}_{t+1}$ must satisfy

$$\boldsymbol{P}_t^{-1} - \sum_{q=1}^{t} \boldsymbol{x}_q \boldsymbol{x}_q^\top - \left( 1 - \frac{a_t}{(1 - a_t) b_t^2} \right) \boldsymbol{x}_{t+1} \boldsymbol{x}_{t+1}^\top \succeq 0.$$

Since

$$1 - \frac{a_t}{(1 - a_t) b_t^2} \geq 0,$$

the Schur complement characterization of symmetric positive semidefinite matrices shows that this is equivalent to the conjunction of

1. $\boldsymbol{P}_t^{-1} - \sum_{q=1}^{t} \boldsymbol{x}_q \boldsymbol{x}_q^\top \succeq 0$,

2. $\boldsymbol{x}_{t+1}$ orthogonal to the kernel of $\boldsymbol{P}_t^{-1} - \sum_{q=1}^{t} \boldsymbol{x}_q \boldsymbol{x}_q^\top$, and

3. $\left( 1 - \frac{a_t}{(1-a_t) b_t^2} \right) \boldsymbol{x}_{t+1}^\top \left( \boldsymbol{P}_t^{-1} - \sum_{q=1}^{t} \boldsymbol{x}_q \boldsymbol{x}_q^\top \right)^\dagger \boldsymbol{x}_{t+1} \leq 1$.

Conditions (1) and (2) are (1) and (2).

Writing $\boldsymbol{x}_{t+1} = c \hat{\boldsymbol{x}}_{t+1}$, we see that

$$b_t^2 = \boldsymbol{x}_{t+1}^\top \boldsymbol{P}_t \boldsymbol{x}_{t+1} = c^2 \hat{\boldsymbol{x}}_{t+1}^\top \boldsymbol{P}_t \hat{\boldsymbol{x}}_{t+1},$$

so Condition (3) is equivalent to

$$\left(1 - \frac{a_t}{(1-a_t)b_t^2}\right)c^2 \le \frac{1}{\hat{\boldsymbol{x}}_{t+1}^\top\left(\boldsymbol{P}_t^{-1} - \sum_{q=1}^t \boldsymbol{x}_q\boldsymbol{x}_q^\top\right)^\dagger \hat{\boldsymbol{x}}_{t+1}}$$

$$\Leftrightarrow \qquad \left(1 - \frac{a_t}{(1-a_t)b_t^2}\right)b_t^2 \le \frac{\hat{\boldsymbol{x}}_{t+1}^\top \boldsymbol{P}_t \hat{\boldsymbol{x}}_{t+1}}{\hat{\boldsymbol{x}}_{t+1}^\top\left(\boldsymbol{P}_t^{-1} - \sum_{q=1}^t \boldsymbol{x}_q\boldsymbol{x}_q^\top\right)^\dagger \hat{\boldsymbol{x}}_{t+1}}$$

$$\Leftrightarrow \qquad \left(1 - \frac{a_t}{(1-a_t)b_t^2}\right)b_t^2 \le d_t(\hat{\boldsymbol{x}}_{t+1}).$$

Finally, it is straightforward to check that the function $\phi$ defined by

$$\phi(b_t^2) := \left(1 - \frac{a_t}{(1-a_t)b_t^2}\right)b_t^2$$

satisfies

$$\phi(b_t^2) = b_t^2 - \frac{\sqrt{4b_t^2 + 1} - 1}{2},$$

and that $\phi(b_t^2) \le \alpha$ iff $b_t^2 \le \alpha + \sqrt{\alpha}$. Combining shows that Condition (3) is equivalent to Condition (3). $\qquad\square$

## G  Calculating the Minimax Directly

As a step in justifying our $\mathcal{ABC}$ assumptions we show that trying to directly calculate the full minimax game is hopeless.

**Lemma 14.** *If we impose the box constraints $|\boldsymbol{x}_T \boldsymbol{P}_T \boldsymbol{s}_{T-1}| \le B_T$ on $\boldsymbol{x}_T$, the first step of the backwards induction evaluates to*

$$\max_{\boldsymbol{x}_T}\min_{\hat{y}_T}\max_{y_T} \boldsymbol{s}_T^\top \boldsymbol{P}_T \boldsymbol{s}_T - \sigma_T^2$$

$$= \boldsymbol{s}_{T-1}^\top \Pi_{T-1}^\dagger \boldsymbol{s}_{T-1} - \sigma_{T-1}^2$$

$$+ \begin{cases} \alpha_T^* \dfrac{\boldsymbol{s}_{T-1}^\top\Pi_{T-1}^\dagger\boldsymbol{s}_{T-1} + \left(\boldsymbol{s}_{T-1}^\top\Pi_{T-1}^\dagger\boldsymbol{s}_{T-1} + B_T^2\alpha_T^*\right)\left(1 - (\alpha_T^*)^2\boldsymbol{s}_{T-1}^\top\Pi_{T-1}^\dagger\boldsymbol{s}_{T-1}\right)}{\left(1 - (\alpha_T^*)^2\boldsymbol{s}_{T-1}^\top\Pi_{T-1}^\dagger\boldsymbol{s}_{T-1}\right)^2} & \text{if } \Pi_{T-1} \text{ is full rank} \\[2em] \max\left\{B_T^2, \alpha_T^* \dfrac{\boldsymbol{s}_{T-1}^\top\Pi_{T-1}^\dagger\boldsymbol{s}_{T-1} + \left(\boldsymbol{s}_{T-1}^\top\Pi_{T-1}^\dagger\boldsymbol{s}_{T-1} + B_T^2\alpha_T^*\right)\left(1 - (\alpha_T^*)^2\boldsymbol{s}_{T-1}^\top\Pi_{T-1}^\dagger\boldsymbol{s}_{T-1}\right)}{\left(1 - (\alpha_T^*)^2\boldsymbol{s}_{T-1}^\top\Pi_{T-1}^\dagger\boldsymbol{s}_{T-1}\right)^2}\right\} & \text{otherwise.}\end{cases}$$

This lemma makes the point that the full minimax formulation leads to an intractable backwards induction, even from the first step.

*Proof.* We prove this lemma by direct calculation. For a given $\boldsymbol{x}_T$, we have already evaluated the $\min_{\hat{y}_T}\max_{y_T}$ argument using the backwards induction under the condition that $|\boldsymbol{x}_T^\top \boldsymbol{P}_T \boldsymbol{s}_{T-1}| \le B_T$. Hence, the above quantity is equal to

$$\max_{\boldsymbol{x}_T} \boldsymbol{s}_{T-1}^\top \boldsymbol{P}_{T-1}\boldsymbol{s}_{T-1} - \sigma_{T-1}^2 + B_T^2 \boldsymbol{x}_T \boldsymbol{P}_T \boldsymbol{x}_T. \tag{14}$$

Next, we extract the $\boldsymbol{x}_T$ dependence from $\boldsymbol{P}_T$. Using $\boldsymbol{P}_{T-1} = \boldsymbol{P}_T + \boldsymbol{P}_T\boldsymbol{x}_T\boldsymbol{x}_T^\top\boldsymbol{P}_T$ and $\boldsymbol{P}_T = (\Pi_{T-1} + \boldsymbol{x}_T\boldsymbol{x}_T)^\dagger$, we have

$$\boldsymbol{P}_{T-1} = \boldsymbol{P}_T + \boldsymbol{P}_T\boldsymbol{x}_T\boldsymbol{x}_T^\top\boldsymbol{P}_T$$
$$= (\Pi_{T-1} + \boldsymbol{x}_T\boldsymbol{x}_T)^\dagger + (\Pi_{T-1} + \boldsymbol{x}_T\boldsymbol{x}_T)^\dagger \boldsymbol{x}_T\boldsymbol{x}_T^\top (\Pi_{T-1} + \boldsymbol{x}_T\boldsymbol{x}_T)^\dagger,$$

and plugging into (14) yields

$$\max_{\boldsymbol{x}_T} \boldsymbol{s}_{T-1}^\top (\Pi_{T-1} + \boldsymbol{x}_T\boldsymbol{x}_T)^\dagger \boldsymbol{s}_{T-1} + \left(\boldsymbol{s}_{T-1}^\top (\Pi_{T-1} + \boldsymbol{x}_T\boldsymbol{x}_T)^\dagger \boldsymbol{x}_T\right)^2$$

$$+ B_T^2\boldsymbol{x}_T (\Pi_{T-1} + \boldsymbol{x}_T\boldsymbol{x}_T)^\dagger \boldsymbol{x}_T - \sigma_{T-1}^2.$$

We need to proceed by cases. First, assume that $\Pi_{T-1}$ is full rank. This implies that $\boldsymbol{x}_T \in \mathcal{R}(\Pi_{T-1})$ and we can apply the second case of Lemma 9 to (14) and arrive at

$$\max_{\boldsymbol{x}_T} \boldsymbol{s}_{T-1}^\top \Pi_{T-1}^\dagger \boldsymbol{s}_{T-1} + \frac{\left(\boldsymbol{s}_{T-1}^\top \Pi_{T-1}^\dagger \boldsymbol{x}_T\right)^2}{1 - \boldsymbol{x}_T^\top \Pi_{T-1}^\dagger \boldsymbol{x}_T} + \left(\boldsymbol{s}_{T-1}^\top \Pi_{T-1}^\dagger \boldsymbol{x}_T + \frac{\boldsymbol{s}_{T-1}^\top \Pi_{T-1}^\dagger \boldsymbol{x}_T \boldsymbol{x}_T^\top \Pi_{T-1}^\dagger \boldsymbol{x}_T}{1 - \boldsymbol{x}_T^\top \Pi_{T-1}^\dagger \boldsymbol{x}_T}\right)^2$$

$$+ B_T^2 \left(\boldsymbol{x}_T \Pi_{T-1}^\dagger \boldsymbol{x}_T + \frac{\left(\boldsymbol{x}_T^\top \Pi_{T-1}^\dagger \boldsymbol{x}_T\right)^2}{1 - \boldsymbol{x}_T^\top \Pi_{T-1}^\dagger \boldsymbol{x}_T}\right) - \sigma_{T-1}^2.$$

Since we assumed that $\boldsymbol{x}_T \in \mathcal{R}(\Pi_{T-1})$ and $\boldsymbol{s}_{T-1} \in \mathcal{R}(\Pi_{T-1})$ by its definition, it is without loss of generality to reparameterize the problem with $\boldsymbol{v} = \left(\Pi_{T-1}^\dagger\right)^{\frac{1}{2}} \boldsymbol{x}_T$ and $\boldsymbol{w} = \left(\Pi_{T-1}^\dagger\right)^{\frac{1}{2}} \boldsymbol{s}_{T-1}$ to obtain

$$\max_{\boldsymbol{v}} \boldsymbol{w}^\top \boldsymbol{w} + \frac{(\boldsymbol{w}^\top \boldsymbol{v})^2}{1 - \boldsymbol{v}^\top \boldsymbol{v}} + \left(\boldsymbol{w}^\top \boldsymbol{v} + \frac{\boldsymbol{v}^\top \boldsymbol{w} \boldsymbol{v}^\top \boldsymbol{v}}{1 - \boldsymbol{v}^\top \boldsymbol{v}}\right)^2 + B_T^2 \boldsymbol{v}^\top \boldsymbol{v} + B_T^2 \frac{(\boldsymbol{v}^\top \boldsymbol{v})^2}{1 - \boldsymbol{v}^\top \boldsymbol{v}} - \sigma_{T-1}^2$$

$$= \max_{\boldsymbol{v}} \boldsymbol{w}^\top \boldsymbol{w} + \frac{(\boldsymbol{w}^\top \boldsymbol{v})^2}{1 - \boldsymbol{v}^\top \boldsymbol{v}} \left(1 + \frac{1}{1 - \boldsymbol{v}^\top \boldsymbol{v}}\right) + B_T^2 \frac{\boldsymbol{v}^\top \boldsymbol{v}}{1 - \boldsymbol{v}^\top \boldsymbol{v}} - \sigma_{T-1}^2.$$

The objective is direction independent except for the $\boldsymbol{w}^\top \boldsymbol{v}$ term, which implies that we should set $\boldsymbol{v} = \alpha \boldsymbol{w}$ for some positive $\alpha$. Plugging in this value of $\boldsymbol{v}$, the optimization problem becomes finding $\alpha^*$, where

$$\alpha^* = \operatorname*{argmax}_{\alpha \geq 0} \ \alpha \left(\boldsymbol{w}^\top \boldsymbol{w}\right)^2 \frac{2 - \alpha^2 \boldsymbol{w}^\top \boldsymbol{w}}{(1 - \alpha^2 \boldsymbol{w}^\top \boldsymbol{w})^2} + B_T^2 \frac{\alpha^2 \boldsymbol{w}^\top \boldsymbol{w}}{1 - \alpha^2 \boldsymbol{w}^\top \boldsymbol{w}}$$

$$= \operatorname*{argmax}_{\alpha \geq 0} \ \alpha \boldsymbol{w}^\top \boldsymbol{w} \frac{\boldsymbol{w}^\top \boldsymbol{w} + (\boldsymbol{w}^\top \boldsymbol{w} + B_T^2 \alpha)(1 - \alpha^2 \boldsymbol{w}^\top \boldsymbol{w})}{(1 - \alpha^2 \boldsymbol{w}^\top \boldsymbol{w})^2}.$$

The objective goes to infinity as $\alpha \to (\boldsymbol{w}^\top \boldsymbol{w})^{-\frac{1}{2}}$, but fortunately the box constraints keep it bounded. The box condition is equivalent to

$$|\boldsymbol{x}_T^\top \boldsymbol{P}_T \boldsymbol{s}_{T-1}| \leq B_T$$

$$\Leftrightarrow \left|\frac{\boldsymbol{x}_T^\top \Pi_T^\dagger \boldsymbol{s}_{T-1}}{1 - \boldsymbol{x}_T^\top \Pi_T^\dagger \boldsymbol{s}_{T-1}}\right| \leq B_T$$

$$\Leftrightarrow \left|\frac{\alpha \boldsymbol{w}^\top \boldsymbol{w}}{1 - \alpha^2 \boldsymbol{w}^\top \boldsymbol{w}}\right| \leq B_T.$$

The left hand side is an increasing function of $\alpha$ and the inequality is satisfied for $\alpha = 0$, and hence the inequality is satisfied for all $\alpha < \alpha_{\max}$, where

$$\alpha_{\max} = \sqrt{1 + \frac{4B_T}{\boldsymbol{w}^\top \boldsymbol{w}}} - 1,$$

the solution to $\alpha \boldsymbol{w}^\top \boldsymbol{w} = (1 - \alpha^2 \boldsymbol{w}^\top \boldsymbol{w}) B_T$. Importantly, this in inequality implies that $1 - \alpha^2 \boldsymbol{w}^\top \boldsymbol{w}$ is bounded below, and hence the maximizer for $\alpha^*$ is well defined.

Hence, we have shown that, in the case when $\boldsymbol{x}_T \in \mathcal{R}(\Pi_{T-1})$,

$$\max_{\boldsymbol{x}_T} \min_{\hat{y}_T} \max_{y_T} \boldsymbol{s}_T^\top \boldsymbol{P}_T \boldsymbol{s}_T - \sigma_T^2$$

$$= \boldsymbol{s}_{T-1}^\top \Pi_{T-1}^\dagger \boldsymbol{s}_{T-1} \left(1 + \alpha_T^* \frac{\boldsymbol{s}_{T-1}^\top \Pi_{T-1}^\dagger \boldsymbol{s}_{T-1} + \left(\boldsymbol{s}_{T-1}^\top \Pi_{T-1}^\dagger \boldsymbol{s}_{T-1} + B_T^2 \alpha_T^*\right)\left(1 - (\alpha_T^*)^2 \boldsymbol{s}_{T-1}^\top \Pi_{T-1}^\dagger \boldsymbol{s}_{T-1}\right)}{\left(1 - (\alpha_T^*)^2 \boldsymbol{s}_{T-1}^\top \Pi_{T-1}^\dagger \boldsymbol{s}_{T-1}\right)^2}\right)$$

$$- \sigma_{T-1}^2,$$

where $\alpha_T^*$, a function of $\boldsymbol{s}_{T-1}^\top \Pi_{T-1}^\dagger \boldsymbol{s}_{T-1}$ and $B_T$, is

$$\underset{0 \le \alpha \le \alpha_{\max}}{\operatorname{argmin}} \; \alpha \frac{\boldsymbol{s}_{T-1}^\top \Pi_{T-1}^\dagger \boldsymbol{s}_{T-1} + (\boldsymbol{s}_{T-1}^\top \Pi_{T-1}^\dagger \boldsymbol{s}_{T-1} + B_T^2 \alpha)(1 - \alpha^2 \boldsymbol{s}_{T-1}^\top \Pi_{T-1}^\dagger \boldsymbol{s}_{T-1})}{\left(1 - \alpha^2 \boldsymbol{s}_{T-1}^\top \Pi_{T-1}^\dagger \boldsymbol{s}_{T-1}\right)^2}$$

for $\alpha_{\max} = \sqrt{1 + \frac{4B_T}{\boldsymbol{s}_{T-1}^\top \Pi_{T-1}^\dagger \boldsymbol{s}_{T-1}}} - 1$.

In the case when $\Pi_{T-1}$ is not full rank, the adversary has the option to play $\boldsymbol{x}_T \notin \mathcal{R}(\Pi_{T-1})$. In this case, applying Lemma 10 to every term yields

$$\max_{\boldsymbol{x}_T} \min_{\hat{y}_T} \max_{y_T} \boldsymbol{s}_T^\top \boldsymbol{P}_T \boldsymbol{s}_T - \sigma_T^2 = \boldsymbol{s}_{T-1}^\top \Pi_{T-1}^\dagger \boldsymbol{s}_{T-1} + B_T^2 - \sigma_{T-1}^2$$

for any $\boldsymbol{x}_T \notin \mathcal{R}(\Pi_{T-1})$ with $|\boldsymbol{x}_T \boldsymbol{P}_T \boldsymbol{s}_{T-1}| \le B_T$.

All in all, the backwards induction applied to the last round yields

$$\max_{\boldsymbol{x}_T} \min_{\hat{y}_T} \max_{y_T} \boldsymbol{s}_T^\top \boldsymbol{P}_T \boldsymbol{s}_T - \sigma_T^2 = \boldsymbol{s}_{T-1}^\top \Pi_{T-1}^\dagger \boldsymbol{s}_{T-1} - \sigma_{T-1}^2 + G$$

where

$$G = \begin{cases} \alpha_T^* \dfrac{\boldsymbol{s}_{T-1}^\top \Pi_{T-1}^\dagger \boldsymbol{s}_{T-1} + \left(\boldsymbol{s}_{T-1}^\top \Pi_{T-1}^\dagger \boldsymbol{s}_{T-1} + B_T^2 \alpha_T^*\right)\left(1 - (\alpha_T^*)^2 \boldsymbol{s}_{T-1}^\top \Pi_{T-1}^\dagger \boldsymbol{s}_{T-1}\right)}{\left(1 - (\alpha_T^*)^2 \boldsymbol{s}_{T-1}^\top \Pi_{T-1}^\dagger \boldsymbol{s}_{T-1}\right)^2} & \text{if } \Pi_{T-1} \text{ is full rank} \\[2em] \max\left\{ B_T^2, \alpha_T^* \dfrac{\boldsymbol{s}_{T-1}^\top \Pi_{T-1}^\dagger \boldsymbol{s}_{T-1} + \left(\boldsymbol{s}_{T-1}^\top \Pi_{T-1}^\dagger \boldsymbol{s}_{T-1} + B_T^2 \alpha_T^*\right)\left(1 - (\alpha_T^*)^2 \boldsymbol{s}_{T-1}^\top \Pi_{T-1}^\dagger \boldsymbol{s}_{T-1}\right)}{\left(1 - (\alpha_T^*)^2 \boldsymbol{s}_{T-1}^\top \Pi_{T-1}^\dagger \boldsymbol{s}_{T-1}\right)^2} \right\} & \text{otherwise.} \end{cases}$$

$\square$

We observe that, in the second case, the maximum could be either term, corresponding to the adversary playing $\boldsymbol{x}_T \notin \mathcal{R}(\Pi_{T-1})$ or $\boldsymbol{x}_T = \alpha_T^* \left(\Pi_{T-1}^\dagger\right)^{\frac{1}{2}} \boldsymbol{s}_{T-1}$, respectively. However, in the later case, the value function obviously ceases to be quadratic and the next step of the backwards induction does in intractable.