[Reviews · NeurIPS 2018]

Reviewer 1



The problem of online linear regression is considered from an individual sequence perspective, where the aim is to control the square loss predictive regret with respect to the best linear predictor $\theta^\top x_t$ simultaneously for every sequence of covariate vectors $x_t \in R^d$ and outcomes $y_t \in R$ in some constraint set. This is naturally formulated as a sequential game between the forecaster and an adversarial environment. In previous work [1], this problem was addressed in the "fixed-design" case, where the horizon T and the sequence of covariate vectors $x_1^T$ is known in advance. The exact minimax strategy (MMS) was introduced and shown to be minimax optimal under natural constraint sets on the label sequence (such as ellipse-constrained labels). The MMS strategy consists in some form of least squares, but where the inverse cumulative covariance matrix \Pi_t^{-1} is replaced by a shrunk version P_t that takes future instance into account. The precision matrices P_t are determined by a backwards recursion (Eq (1)), starting from P_T = \Pi_T^{-1}. This work extends these results to the case when the horizon T and the covariate sequence x_t are unknown a priori, although the latter has to follow some constraint on its cumulative covariance. The key point of the paper is that the backwards recursion can be inverted into a forwards recursion. This enables to play MMS strategy with only the knowledge of the initial precision matrix P_0. The resulting strategy guarantees a regret bound that holds, and is exactly minimax optimal, for the class of all sequences of covariates x_1^T (of any horizon T) for which the backwards MMS strategy would give an initial precision matrix P_0 that coincides with the one chosen by the algorithm (with natural additional constraints on the labels), as shown in Theorem 2. In Theorem 3, it is shown that forward MMS is still minimax when relaxing the above equality on precision matrices into an upper bound, with some necessary additional restrictions on the sequences of covariates and outcomes. Finally, the forward MMS strategy is interpreted as a Follow-the-regularized-leader with a data-dependent regularizer, and Theorem 5 gives an explicit upper bound on minimax regret. The results in this paper constitute a surprising twist on and a significant extension of the results in [1]. The key fact is the ability to invert the backwards recursion, from which the forward "arbitrary horizon/design" algorithm and its minimax optimality follow from results in [1]; this means the results are somewhat incremental, although interesting since they rely on a non-trivial observation. The paper is well-written and clearly motivated, although it has some typos and minor notational inconsistencies which could be fixed. I would rate this a clear accept, if it weren't for one issue with the presentation of the results. Specifically, it is claimed several times that the algorithm is minimax simultaneously for all time horizons T, which can seem quite surprising. While this is true, this comes from the fact that in this setting the scale parameter (a discounted cumulative covariance of the covariate sequence) replaces the time horizon T as a measure of complexity/length of the sequence. There is in fact a correspondence between the two quantities, since the game ends precisely when this scale condition is violated. Hence, while the algorithm is minimax for all horizons T, it is only minimax for one scale parameter, and the universality over T merely comes from the fact that T is no longer used to parameterize the class. This is not as strong as saying that the strategy is minimax simultaneously for all class parameters (here, time horizons), which is the case for the NML algorithm in the log-loss case under restrictive conditions (which is mentioned in lines 53-57). Again, this is very reasonable and the opposite would indeed have been surprising (since the paper considers exact minimax regret, a price would be expected for adaptation to the scale/horizon), so my concern is not with the results per se, but rather with the wording and presentation which could make the above distinction more explicit. As a related remark/question, perhaps the correspondence between \Sigma and T in typical scenarios (e.g., iid sequences) could be somewhat quantified: it would seem that \Sigma scales as (1+log T) C, C being the average/expected covariance matrix. Minor remarks, typos and suggestions: - Lines 20 and 25: seemingly a problem on the font for \mathbb{R}, and an extra ";" in the formula l25; - Eq (MMS): s_t is undefined at this stage; - L46 "allow", l51 "motivate"; - L104: it seems V should also depend on Pi_t; also, perhaps it could be expressed explicitly in terms of its arguments (s_t, etc.); - L112-114: the notations for the constraint sets (in particular their arguments) seem to vary a lot in the text (shouldn't B depend on x_t ?); see also line 135; - L158: maybe it could be useful to say how the P_s are defined in this sentence (with forward recursion starting with Sigma_0); - L187: the union should be an intersection; - Related to the aforementioned point, maybe the dependence on the initial precision matrix of forward MMS could be made explicit in the notations for clarity; - There is no conclusion at the end of the paper, likely due to space constraints, hopefully with the additional page in camera-ready some perspectives for future work will be included; - Some missing references in lines 453 and 479. [1] Peter L. Bartlett, Wouter M. Koolen, Alan Malek, Manfred K. Warmuth, and Eiji Takimoto. Minimax fixed-design linear regression. In Proceedings of The 28th Annual Conference on Learning Theory (COLT), pages 226–239, 2015. EDIT: I read the authors' response, which addressed the issue on the presentation of the results. The updated formulation is more convincing and accurately describes the content of the results. I have updated my score accordingly. On another note, since the scale parameter $\Sigma_0$ is hard to interpret as is, I still think it could be interesting to relate it to more familiar parameters like the time horizon T and some sort of average covariance matrix, if only in some restricted case. This could give some sense of when the algorithm should be expected to stop, or when its minimax regret guarantee starts to become "tight", or what the cost of "misspecification" (i.e., anticipating a wrong covariance matrix via the choice of $\Sigma_0$) is.

Reviewer 2



This paper develops a theory for minimax linear regression in online setting, by extending the previous study by Bartlett et al. (2015). Overall, the paper is clearly written and the result is appealing. - The interpretation as a follow-the-regularized leader strategy is interesting. It would be more interesting if this interpretation (in particular, the form of regularization) can be explained intuitively. - Is it difficult to extend the results to other loss functions? - Is it possible to generalize the results to other regression models such as logistic regression? If not, where does the difficulty arise?

Reviewer 3



Authors demonstrate a forward recursion (interpretable as a a data-dependent regularized follow-the-leader) is minimax optimal for linear regression given (scale) constraints on the adversary play. Prior work established the minimax strategy (mms) given the complete sequence of covariates via a backwards recursion. This work first shows that summarizing the complete sequence of covariates by an initial second-moment-style equality constraint is sufficient to compute the mms via a forward recursion. Then, the equality constraint is relaxed to an inequality constraint with additional conditions that ensure that the inequality constraint will be active given optimal adversarial play. I have not verified any of the proofs.